# Unlocking SVD-Space for Feedback Aligned Local Training

## Abstract

Deep Neural Networks (DNNs) are typically trained using backpropagation, which, despite its effectiveness, requires substantial memory and computing resources. To address these limitations, we propose a novel local training framework that enables efficient and scalable neural network training without relying on global backpropagation. Our framework harnesses the alignment of Singular Value Decomposed (SVD) weight space with feedback matrices, guided by custom layerwise loss functions, to enable efficient and scalable neural network training. We decompose weight matrices into their SVD components before training, and perform local updates on the SVD components themselves, driven by a tailored objective that integrates feedback error, alignment regularization, orthogonality constraints, and sparsity. Our approach leverages Direct Feedback Alignment (DFA) to eliminate the need for global backpropagation and further optimizes model complexity by dynamically reducing the rank of the SVD components during training. The result is a compute- and memory-efficient model with classification accuracy on par with traditional backpropagation while achieving a 50-75% reduction in memory usage and computational cost during training. With strong theoretical convergence guarantees, we demonstrate that training in the SVD space with DFA not only accelerates computation but also offers a powerful, energy-efficient solution for scalable deep learning in resource-constrained environments. Code is available.

## 1 Introduction

As neural networks grow in size and complexity, the memory and computational requirements for training have become a significant bottleneck, particularly in resource-constrained environments such as edge devices. Backpropagation (BP), the most commonly used method for training deep networks, increases this issue as it relies on a global loss objective and needs to store intermediate activations for layer-by-layer gradient updates. This not only demands high memory usage and computational power, but also introduces the update-locking issue(Lillicrap et al., 2020), in which parameters of the hidden layer cannot be updated until both forward and backward computations are completed. Hence, we highlight the need for alternative training paradigms that are both more memory and compute-efficient and enable efficient parallelization of the training process.

Direct Feedback Alignment (DFA) (Nøkland, 2016) offers a promising alternative to backpropagation (BP) by enabling local layerwise updates without global gradient propagation, which reduces the hardware complexity of neural network training. This makes DFA appealing for resource-constrained environments. However, DFA faces limitations in scaling to more complex tasks and deeper architectures, largely due to its reliance on fixed random feedback connections. These connections limit its ability to train deep convolutional layers and perform well on datasets like CIFAR-100 and ImageNet without transfer learning (Launay et al., 2020). Although the integration of the Kolen-Pollack algorithm (Akrout et al., 2019) has improved feedback alignment by learning symmetric feedback weights, DFA still struggles to match BP in terms of accuracy and scalability. While DFA's simplified learning rules have been well-studied, its potential combination with low-rank subspaces—such as those from Singular Value Decomposition (SVD)—remains underexplored. This synergy could be critical, as low-rank subspaces can reduce computational complexity, complementing DFA's goal of simpler learning rules.

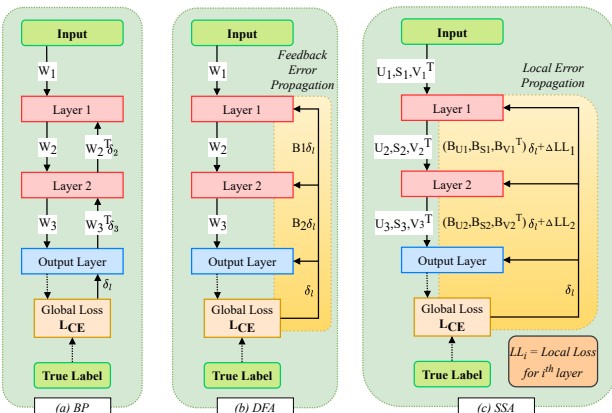

Figure 1: A comparison of neural network training methods. Notations: $W$ = forward weights, $Layer$ = Layer Activations, $L_{CE}$ = cross-entropy loss, $B$ = random feedback weights, $\delta$ = gradients, $\delta_l$ = gradient of $L_{CE}$ = feedback error, $U, S, V^T$ = SVD components of forward weights, $B_U, B_S, B_{V^T}$ = SVD components of feedback weights, Local Loss (LL) has components: $L_{align}$ = Alignment loss, $L_{cos}$ = Cosine similarity loss, $L_{ortho}$ = Singular Vector Orthogonality regularizer, $L_{Hoyer}$ = Hoyer regularizer. **(a) Backpropagation (BP):** Global gradient updates through each layer. **(b) Direct Feedback Alignment (DFA):** Local updates with random feedback **(c) SVD-space Alignment (SSA) (ours):** Decomposes weights into SVD components before training and aligns feedback on those decomposed components itself with local losses. Note: LL + $L_{CE}$ is the total local loss objective in our work.

We propose a novel hybrid training framework- SVD-Space Alignment (SSA) that synergistically combines the simplified local learning rules of Direct Feedback Alignment (DFA) with the structured learning advantages of Singular Value Decomposition (SVD). SVD decomposes weight matrices into orthogonal components, allowing for efficient updates in low-rank subspaces. By aligning the SVD spaces of both the forward weights and feedback matrices, our method introduces structure into the learning process, countering the randomness inherent in DFA's feedback matrices and improving scalability, stability, and faster convergence. Our approach begins by decomposing the weight matrices into their SVD components— $U$, $S$, and $V^T$ —before training commences. We apply local updates to the SVD components using a custom loss function that minimizes model cross-entropy loss, maintains alignment between forward and feedback weights, preserves orthogonality, and enforces sparsity for efficiency. A key feature is the progressive rank reduction of SVD components during training, reducing memory and computation while enabling energy-efficient inference optimized for resource-constrained environments. Our work introduces several key innovations that make DFA suitable for deep convolutional layers, addressing a significant gap in prior research. Previous attempts to apply local loss methods to convolutional layers have relied on flattening these layers, which leads to a loss of hierarchical and spatial features critical for vision-based applications. In contrast, we preserve the convolutional structure by applying spatial-wise decomposition embedded within SVD matrices, allowing our method to retain the hierarchical information necessary for accurate visual recognition tasks.

Our key contributions are summarized as follows:

- We propose a novel hybrid training framework that incorporates feedback alignment with Singular Value Decomposition (SVD), leveraging structured updates in SVD-space to mitigate the limitations of random feedback.

- We introduce a custom loss function that incorporates feedback error, alignment loss, orthogonality regularization, and sparsity constraints. This ensures that local layerwise updates in the SVD-space are efficient and convergent.

- We demonstrate significant reductions in memory usage and computational complexity through progressive rank reduction, achieving up to 50-75% reductions in memory usage and computational requirements.

- We show that our method can successfully train deep convolutional networks while retaining spatial and hierarchical information. We provide theoretical convergence guarantees and empirical validation, showing that our method achieves classification accuracy on par with backpropagation, with faster convergence and lower computational costs, on challenging datasets like CIFAR-100 and ImageNet.

## 2 RELATED WORK

Our work draws upon several lines of research, including Direct Feedback Alignment (DFA), the application of Singular Value Decomposition (SVD) in neural networks, local layerwise training methods, and model compression techniques. This section reviews these areas.

**Direct Feedback Alignment (DFA):** Direct Feedback Alignment (DFA) (Nøkland, 2016) was introduced as a biologically plausible alternative to backpropagation (BP), reducing memory usage by decoupling weight updates from global error gradients and using random feedback connections. However, DFA struggles with gradient alignment, leading to instability in deeper networks. Nøkland and Eidnes (Nøkland & Eidnes, 2019) improved DFA by introducing local error signals and auxiliary classifiers, enabling more stable layer-wise updates in deep networks. Building on this, our approach aligns SVD-decomposed weight spaces with feedback matrices, using custom loss functions to reduce feedback randomness and improve DFA's scalability, stability, and efficiency, while lowering model complexity.

**Singular Value Decomposition (SVD) in Neural Networks:** Singular Value Decomposition (SVD) is widely used in machine learning to compress models and improve computational efficiency by reducing the dimensionality of weight matrices. Early works, like (Denil et al., 2013), showed that low-rank approximations can lower the number of parameters and FLOPs with minimal performance loss. More recent approaches, such as (Denton et al., 2014) and (Yang et al., 2020), applied SVD to CNNs, achieving significant inference speedups. Although (Yang et al., 2020) also incorporated an orthogonality regularizer to maintain stability, these methods still rely on backpropagation for updates, which can disrupt orthogonality. Our method addresses this by integrating SVD with Direct Feedback Alignment (DFA), preserving orthogonality and dynamically reducing the rank during training to create compact models without sacrificing accuracy.

**Local Layerwise Training:** Local learning rules, which update layers independently of global gradients, are gaining attention for their scalability and efficiency. Methods like greedy layerwise training (Belilovsky et al., 2019), DRTP (Frenkel et al., 2021), and AugLocal (Ma et al., 2024), reduce backpropagation's memory costs using local losses or auxiliary networks. Forward-only methods, such as Hinton's Forward-Forward (FF) (Hinton, 2022) and PEPITA (Dellaferrera & Kreiman, 2022), avoid backprop entirely but often fall short in accuracy, especially in deeper networks, and struggle with global alignment. Our work focuses on improving local loss methods, as forward-only approaches struggle with classification accuracy in deeper networks. We enhance local layerwise training by applying Singular Value Decomposition (SVD) to layer weights and aligning components with feedback matrices using Direct Feedback Alignment (DFA).

**Model Compression Techniques:** Model compression techniques like pruning, quantization, and low-rank factorization (Marinó et al., 2023) reduce neural network size and computation but often require retraining to recover accuracy and rely on backpropagation. Our approach trains low-rank SVD components from scratch, with no retraining needed. Progressive rank reduction dynamically adjusts model complexity, maintaining accuracy while creating compact models. Using DFA for feedback also improves hardware efficiency and scalability.

## 3 METHODOLOGY

We present SVD-Space Alignment (SSA), a novel method combining Singular Value Decomposition (SVD) with Direct Feedback Alignment (DFA) for efficient local layerwise training in deep neural networks. By decomposing weight matrices with SVD and applying local updates using our custom loss function (including DFA's feedback error), SSA reduces computational complexity and memory usage. The custom loss function ensures structured learning, incorporating alignment, orthogonality, and sparsity regularization. Additionally, a dynamic rank reduction strategy further optimizes the model for edge devices.

### 3.1 SVD-Space Decomposition

Given a weight matrix $W_i \in \mathbb{R}^{m \times n}$ for layer $i$, we decompose it into its SVD form:

$$W_i = U_i S_i V_i^T \tag{1}$$

where $U_i \in \mathbb{R}^{m \times r}$ and $V_i \in \mathbb{R}^{n \times r}$ are orthogonal matrices, and $S_i \in \mathbb{R}^{r \times r}$ is a diagonal matrix of singular values, with $r \leq \min(m, n)$. Decomposing weights before training allows updates to be made directly in the SVD-space, preserving the orthogonality of $U_i$ and $V_i^T$ while enabling efficient rank reduction during training. This avoids the computational overhead of performing SVD at every epoch and promotes more structured learning by preventing orthogonality disruption caused by gradient descent in BP.

**SVD-Space for Convolutional Layers.** For convolutional layers, we integrate the SVD-space approach by leveraging spatial decomposition inspired by (Yang et al., 2020). A convolutional kernel $K \in \mathbb{R}^{N \times C \times H \times W}$, with $N$ as the number of filters, $C$ as the number of input channels, and $H \times W$ as the spatial dimensions, is first reshaped into a 2D matrix $K' \in \mathbb{R}^{NW \times CH}$. This matrix is then decomposed using SVD:

$$K' = U\Sigma V^T \tag{2}$$

where $U \in \mathbb{R}^{NW \times r}$ and $V \in \mathbb{R}^{CH \times r}$ are unitary matrices, and $\Sigma \in \mathbb{R}^{r \times r}$ is a diagonal matrix of singular values, with $r = \min(NW, CH)$.

The decomposed components are reshaped back into convolutional layers as follows:

- $U\sqrt{\Sigma}$ is reshaped into a convolutional kernel $K_1 \in \mathbb{R}^{r \times C \times H \times 1}$,
- $\sqrt{\Sigma}V^T$ is reshaped into a kernel $K_2 \in \mathbb{R}^{N \times r \times 1 \times W}$.

This decomposition splits the original convolutional operation into two consecutive layers, preserving hierarchical and spatial features while reducing computational complexity. During the forward pass, the convolutions are performed using $K_1$ and $K_2$. During backpropagation, gradients are computed for $K_1$ and $K_2$ and used to directly update these decomposed kernels without reconstructing the original kernel $K$. We retain the decomposed kernels for inference.

### 3.2 Direct Feedback Alignment in SVD-Space

DFA uses random feedback matrices that are independent of the inter-layer weight matrices, enabling efficient local updates. For each layer $i$, the feedback matrices $B_{U_i}$, $B_{S_i}$, and $B_{V_i^T}$ are used to update the SVD components $U_i$, $S_i$, and $V_i^T$, respectively. The update rules for each SVD component are defined as follows:

$$U_i^{(t+1)} = U_i^{(t)} - \eta \nabla_{U_i} L_i(U_i, S_i, V_i^T) \tag{3}$$

$$S_i^{(t+1)} = S_i^{(t)} - \eta \nabla_{S_i} L_i(U_i, S_i, V_i^T) \tag{4}$$

$$V_i^{T(t+1)} = V_i^{T(t)} - \eta \nabla_{V_i^T} L_i(U_i, S_i, V_i^T) \tag{5}$$

where $\eta$ is the learning rate, and $L_i(U_i, S_i, V_i^T)$ is the layer-specific loss function (detailed in Section 3.3). These updates are independent across layers, reducing the computational overhead typically associated with backpropagation and enabling parallelization of layer updates.

### 3.3 Training Objective: Custom Layerwise Loss Function

We design a custom layerwise loss function with regularization terms to maintain the model's structure and efficiency in the SVD-space. We provide the gradients of this loss to the layer as shown in Fig 1. The local loss function for layer $i$ is formulated as:

$$LL_i(\theta_i) = \alpha L_{\text{CE}}(\theta_i) + \beta L_{\cos}(\theta_i) + \gamma L_{\text{align}}(\theta_i) + \delta L_{\text{ortho}}(\theta_i) + \epsilon L_{\text{Hoyer}}(\theta_i) \tag{6}$$

where $\theta_i = (U_i, S_i, V_i^T)$, and the terms are defined as follows:

**Cross-Entropy Loss** ($L_{\text{CE}}$): This loss is the model cross entropy loss. We get the feedback error, matching DFA's local error, from the derivative of this loss. The feedback error measures how well

the model's predictions align with target outputs, crucial for classification tasks. The feedback error comes from the derivative of the entire model's cross entropy loss.

$$\Delta L_{CE} = y_{predict} - y_{label} \tag{7}$$

**Cosine Similarity Loss** ($L_{\cos}$): This loss encourages alignment between the activations of the network and the feedback signals, ensuring that the directions of the activations remain consistent with the feedback.

$$L_{\cos} = 1 - \frac{\langle \text{layer\_input} \cdot (USV^T), (B_U B_S B_{V^T})^T \cdot e \rangle}{\|\text{layer\_input} \cdot (USV^T)\| \cdot \|(B_U B_S B_{V^T})^T \cdot e\|} \tag{8}$$

Where $layer\_input \cdot (USV^T)$ is the activation from the network. $(B_U B_S B_{V^T})^T \cdot e$ is the feedback signal with $e = \Delta L_{CE}$.

**Alignment Loss** ($L_{\text{align}}$): This loss ensures alignment between the SVD-decomposed matrices ($U_i$, $S_i$, $V_i^T$) and their respective feedback matrices ($B_{U_i}$, $B_{S_i}$, $B_{V_i^T}$). This loss is defined as:

$$L_{\text{align}}(\theta_i) = \|U_i - B_{U_i}\|_F^2 + \|S_i - B_{S_i}\|_F^2 + \|V_i^T - B_{V_i^T}\|_F^2 \tag{9}$$

**Singular Vector Orthogonality Regularizer** ($L_{\text{ortho}}$): This component promotes orthogonality in the singular vectors $U_i$ and $V_i^T$, which preserves the structure of the SVD decomposition. The regularizer is defined as:

$$L_{\text{ortho}}(\theta_i) = \|U_i^T U_i - I\|_F^2 + \|V_i^T V_i - I\|_F^2 \tag{10}$$

**Hoyer Regularizer** ($L_{\text{Hoyer}}$): This regularizer encourages sparsity in the singular values by minimizing the ratio of the $L_1$ norm to the $L_2$ norm of the singular values. We apply this loss every ten epochs to enhance sparsity and reduce the rank. The regularizer is defined as:

$$L_{\text{Hoyer}}(\theta_i) = \frac{\|S_i\|_1}{\|S_i\|_2} \tag{11}$$

Each term in the composite loss function serves a specific purpose in ensuring efficient, structured, and stable learning of the SVD-decomposed weights (detailed in the **Appendix**). By jointly minimizing these objectives, the model is able to achieve efficient training.

### 3.4 DYNAMIC RANK REDUCTION STRATEGY

To reduce model size and computational complexity, we employ a dynamic rank reduction strategy. Initially, each layer's SVD decomposition starts with full rank $r_0$. During the initial epochs, we use an **epoch-based schedule**, progressively reducing the rank every ten epochs by applying the Hoyer regularizer to sparsify the matrices. The rank $r_k$ at epoch $k$ is:

$$r_k = r_0 \times \left(1 - \frac{\lfloor \frac{k}{10} \rfloor}{K/10}\right),$$

where $K$ is the total number of epochs.

In the later epochs, defined dynamically as the point where the rank $r_k$ has reduced to $\zeta r_0$ ($\zeta = 0.7$), we incorporate a **threshold-based check** to dynamically retain singular values contributing to a predefined energy threshold (95% of the matrix's total energy). This mitigates the sharp rank reduction induced by $1 - k/K$ in epoch-based scheduling as $k$ approaches $K$. If the threshold-based check determines that the rank is already sufficiently low from the epoch-based reduction, further reduction is halted to preserve the model's representational capacity.

### 3.5 COMPUTATIONAL AND MEMORY COMPLEXITY

The computational and memory requirements of the SSA method are significantly lower than those of traditional BP, both during training and inference, due to the use of dynamic rank reduction and low-rank SVD representations of the neural network weights.

**Training Complexity.** For a weight matrix $W_i \in \mathbb{R}^{m \times n}$, the computational cost of traditional BP is $O(m \times n \times p)$, where $p$ is the batch size. In SSA, the weight matrix is decomposed into SVD components: $U \in \mathbb{R}^{m \times r}$, $S \in \mathbb{R}^{r \times r}$, and $V^T \in \mathbb{R}^{r \times n}$, where $r \ll \min(m,n)$. The computational cost of updating these components is:

$$O(m \times r \times p) + O(r \times p) + O(r \times n \times p), \tag{12}$$

where $O(m \times r \times p)$: Updating $U$; $O(r \times p)$: Updating the diagonal matrix $S$; $O(r \times n \times p)$: Updating $V^T$. As the rank $r$ is progressively reduced throughout training, the computational cost decreases further, leading to substantial efficiency gains. The memory complexity is also reduced compared to BP. Instead of storing the full weight matrix $W_i$ and its gradients, SSA stores only the SVD components. The resulting memory complexity is:

$$O(m \times r) + O(r \times r) + O(r \times n), \tag{13}$$

which becomes increasingly efficient as $r$ decreases over the course of training.

**Inference Complexity.** During inference, SSA leverages the decomposed form $USV^T$ instead of the full weight matrix $W_i$. The computational complexity for inference is:

$$O(r \times n) + O(r \times r) + O(m \times r), \tag{14}$$

where $O(r \times n)$: Computing $V^T x$ (given $x$ = Layer input), $O(r \times r)$: Scaling by $S$, $O(m \times r)$: Projecting with $U$. Since $r \ll \min(m,n)$, inference is lightweight, ensuring efficiency in computationally constrained environments. The memory requirements during inference are also minimal. Gradients and activations are no longer required, and only the final decomposed components $U$, $S$, and $V^T$ need to be stored, leading to the same memory complexity as training:

$$O(m \times r) + O(r \times r) + O(r \times n). \tag{15}$$

## 4 Experimental Setup

In this section, we describe the experimental setup used to evaluate the performance of our proposed SSA method. We outline the datasets, neural network architectures, baseline methods for comparison, and the evaluation metrics used to assess the effectiveness of our approach. Further details are elaborated in the Appendix.

**Datasets:** We evaluate our method on CIFAR-10, CIFAR-100 (Krizhevsky et al., 2009), and ImageNet (ILSVRC-2012) (Krizhevsky et al., 2017), using CIFAR for small-scale benchmarks and ImageNet for large-scale scalability.

**Network Architectures:** To demonstrate the flexibility of our approach, we evaluate the SSA method on several common neural network architectures: SmallConv (conv96-pool-conv192-pool-conv512-pool-fc1024), VGG-13 (Simonyan & Zisserman, 2014), and ResNet-32 (He et al., 2016). These models cover a range of depths and sizes, allowing us to assess the performance of the SSA method on both small and large networks.

**Baselines:** We compare our SSA method against several baselines to evaluate accuracy, memory, computational cost, and training stability. These include BP, the standard method for neural network training; Direct Feedback Alignment (DFA) (Nøkland, 2016), a biologically plausible and scalable alternative to BP; SVD-BP, which combines low-rank approximation with backpropagation but lacks feedback alignment; PredSim (Nøkland & Eidnes, 2019), which uses local error signals like reconstruction and similarity matching loss; AugLocal (Ma et al., 2024), a local learning approach with augmented auxiliary networks; DRTP (Frenkel et al., 2021), a gradient-free method using Direct Random Target Projection; and PEPITA (Dellaferrera & Kreiman, 2022), which applies error-driven forward-only local learning.

**Evaluation Metrics:** To evaluate the SSA method, we use several key metrics: Classification Accuracy, measuring top-1 and top-5 accuracy to assess how well the model generalizes to unseen data; Memory Usage, tracking the model's memory footprint during training to evaluate efficiency in resource-constrained environments and computational Cost, calculating the reduction in FLOPs per training iteration to gauge computational efficiency. These metrics provide a comprehensive view of SSA's performance in terms of accuracy, efficiency, and scalability compared to baselines.

**Training Details:** All experiments are conducted on a machine with NVIDIA A40 GPUs and 48 GB of GDDR6 memory. We implement the SSA method and baselines using PyTorch. Training is performed with the Adam optimizer (Kingma, 2014), with a learning rate ranging from $1e-4$ to $5e-4$ (detailed in Appendix), and data augmentation techniques like random cropping and horizontal flipping to improve generalization. The batch size is 128 for CIFAR-10/100 and 256 for ImageNet. For SSA, we start with the initial rank $r_0$ at the full rank of the original weight matrices, progressively reducing it as outlined in Section 3.4.

**Hyperparameter Selection.** The coefficients $\alpha, \beta, \gamma, \delta, \epsilon$ are set as $(0.1, 0.01, 0.1, 0.05, 0.01)$, consistent across all experiments. Feedback cross-entropy error and alignment loss, being convex and smooth, are assigned higher weights, while non-convex components such as cosine similarity loss, orthogonality loss, and the Hoyer regularizer have lower weights (Theoretical Analysis in Appendix). To mitigate non-convexity, we ensure quasi-convexity by projecting the cosine similarity loss and SVD component norms onto the unit sphere. The Hoyer regularizer is smoothed (71) and applied only every ten epochs during rank reduction.

Theoretical analysis and ablation studies (Section 5.3) guide these choices. Ablation results demonstrate the impact of removing individual loss components, emphasizing the need to balance their contributions. To confirm the robustness of the coefficients, we perform basic $k = 3$-fold cross-validation, evaluating a small grid of candidate values. This process takes approximately 5%-10% of the total training time and leverages the independence of local loss objectives, avoiding the complexity of global optimization. Once selected, the coefficients remain fixed across all experiments.

## 5 RESULTS AND ANALYSIS

In this section, we present the results of our experiments and provide an in-depth analysis of the performance of the proposed SSA method. We compare the results against baseline methods. Our analysis covers classification accuracy, memory and computational efficiency, convergence rates, ablation studies, and energy efficiency.

### 5.1 CLASSIFICATION ACCURACY

We present the classification accuracies of the SSA method on CIFAR-10 and ImageNet, compared to baseline methods. Table 1 shows a comparison with BP, SVD-BP, local training, and forward-only methods. The Forward-Forward method is excluded as it does not extend to convolutional networks, while DFA, DRTP, and PEPITA are omitted from the next Table 2 due to their inability to scale to larger networks without encountering heavy accuracy loss. We see that our method outperforms other local training and forward-only methods while achieving performance on par with BP.

| Network | Method | CIFAR-10 (mean $\pm$ std) | CIFAR-100 (mean $\pm$ std) |
|---------|--------|---------------------------|----------------------------|
| **SmallConv** | BP | $87.57 \pm 0.14$ | $62.25 \pm 0.21$ |
| | SVD-BP (Yang et al., 2020) | $87.30 \pm 0.18$ | $61.64 \pm 0.19$ |
| | DFA (Akrout et al., 2019) | $73.10 \pm 0.53$ | $44.93 \pm 0.51$ |
| | DRTP (Frenkel et al., 2021) | $68.96 \pm 0.80$ | NA |
| | PEPITA (Dellaferrera & Kreiman, 2022) | $56.34 \pm 1.24$ | $27.56 \pm 0.67$ |
| | SSA (ours) | $86.23 \pm 0.12$ | $60.88 \pm 0.17$ |

Table 1: Comparison of classification accuracy (mean $\pm$ standard deviation) over 5 independent runs with random inits for CIFAR-10 and CIFAR-100 datasets.

In Table 2, we compare our method to BP, SVD-BP, and more recent local layerwise training methods on CIFAR-10 and ImageNet datasets, focusing on larger networks.

From Table 2, we observe that SSA consistently achieves classification accuracy comparable to standard backpropagation (BP) across all datasets when applied to VGG-like networks. On CIFAR-10, SSA's accuracy is within 0.2% of BP, and the gap remains minimal on ImageNet as well. However, for ResNet-32, SSA shows a slightly larger accuracy gap compared to BP, indicating room for improvement in deeper networks. Notably, PredSim does not report statistics for ImageNet in their paper, limiting direct comparison. AugLocal, on the other hand, embeds properties of later layers

| Network | Method | CIFAR10 (Top-1) | ImageNet (Top-1) | ImageNet (Top-5) |
|---------|--------|-----------------|------------------|------------------|
| **VGG-13** | BP | 93.75 | 71.59 | 90.39 |
| | SVD-BP (Yang et al., 2020) | 92.8 | 71.37 | 90.2 |
| | PredSim (Nøkland & Eidnes, 2019) | 86.49 | NA | NA |
| | AugLocal (Ma et al., 2024) | 93.72 | 70.93 | 90.16 |
| | SSA (ours) | 92.7 | 69.87 | 89.7 |
| **ResNet-32** | BP | 93.74 | 74.28 | 91.76 |
| | SVD-BP (Yang et al., 2020) | 91.77 | 72.91 | 89.27 |
| | PredSim (Nøkland & Eidnes, 2019) | 79.31 | NA | NA |
| | AugLocal (Ma et al., 2024) | 93.47 | 73.95 | 91.7 |
| | SSA (ours) | 88.53 | 70.03 | 88.78 |

Table 2: Performance comparison (accuracy %) of various methods on CIFAR-10 and ImageNet datasets for VGG-13 and ResNet-32 architectures.

into earlier layers, effectively aligning with the global loss objective. However, this comes at the cost of increased computational overhead due to the auxiliary networks introduced for each layer.

## 5.2 MEMORY AND COMPUTATIONAL EFFICIENCY

One of the primary advantages of the SSA method is its reduction in memory usage and computational cost due to progressive rank reduction of the SVD components during training process.

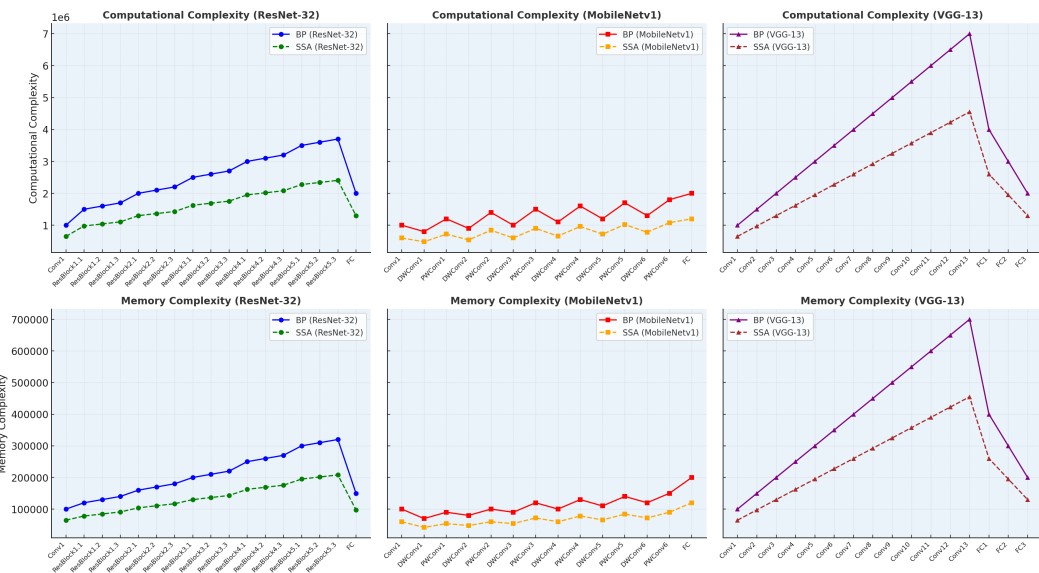

Figure 2: BP and SSA compute and memory per layer for ResNet-32, MobileNetV1 and VGG-13

SSA reduces memory usage by up to 50% compared to backpropagation and reduces compute by at least 40% across various model architectures as demonstrated in Fig 2. We see similar compute-memory savings in inference as well, as explained in Section 3.5. These results show that SSA is particularly suitable for deployment in resource-constrained environments.

## 5.3 ABLATION STUDY

We perform an ablation study to quantify the contribution of each component in the composite loss function to the overall performance of SSA. Table 3 shows the impact of removing each component (Cross-Entropy Loss, Cosine Similarity Loss, Alignment Loss, Orthogonality Regularizer, and Hoyer Regularizer) on classification accuracy and computational efficiency.

From Table 3, removing the Cross-Entropy Loss results in a significant accuracy drop, as expected. Cosine similarity loss attempts to preserve the direction of gradient update proportional to true gra-

Table 3: Ablation study on CIFAR-10 showing the impact of each component in the composite loss function.

| Component Removed | Accuracy | FLOPs (in billions) |
|---|---|---|
| Full SSA (All Components) | 92.7% | 0.14 |
| No Cross-Entropy Loss | 70.5% | 0.133 |
| No Cosine Similarity Loss | 87%% | 0.126 |
| No Alignment Loss | 83.1% | 0.119 |
| No Orthogonality Regularizer | 85.4% | 0.112 |
| No Hoyer Regularizer | 90.5% | 0.105 |

dient of BP. However, as we don't have BP gradients during the training, we approximate true gradient by (layer input x weight) layer output, so that the direction of the update remains consistent. Alignment loss attempts to reduce the loss between the subspaces in forward and feedback weights. Both cosine similarity loss and alignment loss aids gradient direction preservation, and therefore, removing these components also decreases accuracy, but less severely than Cross-Entropy Loss. Orthogonality Regularizer maintains the unitary properties of $U, Vt$. If the unitary properties are maintained, the angular alignment and any angular transformation will be meaningful (preserving lengths and angles). Hence, removing the regularizer negatively impacts both accuracy and computational efficiency, which might worsen in deeper models. The Hoyer regularizer sparsifies weights during rank reduction and has limited effect on the overall accuracy. Overall, the ablation study demonstrates that each loss component is essential for SSA's performance and efficiency.

## 6 DISCUSSION

The experimental results demonstrate that our proposed SSA method achieves competitive classification accuracy while significantly reducing memory usage and computational cost. In the following sub-sections, we state the advantages of SSA over DFA and its limitations.

### 6.1 COMPARISON WITH DFA

SSA introduces two key distinctions from DFA: the use of a structured weight-space (SVD-decomposed weights) and custom loss components applied directly in this SVD-space. To evaluate these differences, we present comparisons between SSA and DFA, along with its variants (Sanfiz & Akrout, 2021), in the following tables and figures.

| Method | LeNet | ResNet-20 | ResNet-56 |
|---|---|---|---|
| BP | 15.92 | 10.01 | 7.83 |
| FA | 40.67 | 29.59 | 29.23 |
| DFA | 37.59 | 32.16 | 32.02 |
| uSF | 16.34 | 10.59 | 9.19 |
| brSF | 17.08 | 11.08 | 10.13 |
| SSA | 16.20 | 10.60 | 9.80 |

Table 4: CIFAR-10 test error (%) for different methods

| Method | Top-1 Error Rate (%) |
|---|---|
| BP | 30.39 |
| FA | 85.25 |
| DFA | 82.45 |
| uSF | 34.97 |
| brSF | 37.21 |
| SSA (Ours) | 32.45 |

Table 5: ImageNet test error rates for ResNet-18

One variant of DFA, known as **Uniform Sign-concordant Feedbacks (uSF)**, generates feedback weights by preserving the sign of the forward weight matrices while assuming unit magnitude for the synaptic weights. This is mathematically represented as $B_i = \text{sign}(W_i^T) \forall i$. Another variant, **Batchwise Random Magnitude Sign-concordant Feedbacks (brSF)**, extends uSF by assigning random magnitudes $|R_i|$ to the feedback weights after each update while

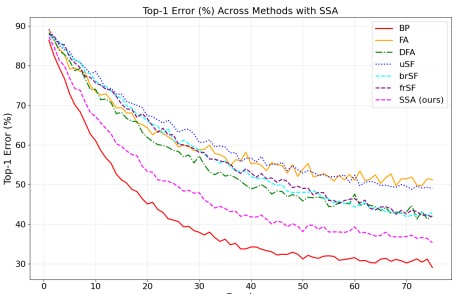

Figure 3: Top-1 Error (%) Across Epochs for a 3-layer MLP.

retaining the sign of the forward weights. The feedback weights in this case are defined as $B_i = |R_i| \cdot sign(W_i^T) \, \forall i$.

We evaluate SSA and DFA (including its variants) on CIFAR-10 and ImageNet datasets. Tables 4 and 5 summarize the test error rates for these methods, including the baseline BP. Our results indicate that SSA outperforms most variants of DFA across both datasets. To further analyze convergence behavior, we plot the error across epochs for a 3-layer MLP trained with SSA, BP, and DFA (including its variants) in Figure 3. The results illustrate that SSA converges significantly faster than DFA and its variants, showcasing the advantages of structured feedback and custom loss design. While DFA and its variants perform poorly on convolutional layers, or cannot be directly applied to them, SSA achieves robust performance across both fully connected and convolutional architectures. To ensure uniformity in comparisons, the plotted results focus on MLPs. This limitation of DFA on convolutional layers further highlights the versatility of SSA for broader network types.

### 6.2 LIMITATIONS

While SSA demonstrates promising results, it also presents certain limitations that warrant further exploration:

**Rank Reduction Trade-offs:** Although the progressive rank reduction strategy effectively reduces memory and computational costs, it may introduce performance trade-offs, especially in scenarios where an aggressive reduction in rank leads to a loss of model capacity. In some cases, the reduced representational power could result in lower accuracy, particularly for highly complex tasks or datasets. This suggests the need for careful tuning of the rank reduction schedule, potentially adapting it dynamically based on the task's complexity or during different training phases.

**Hyperparameter Sensitivity:** The performance of the SSA method is sensitive to the choice of hyperparameters, particularly the coefficients $(\alpha, \beta, \gamma, \delta, \epsilon)$ that weigh the individual loss components in the composite loss function. While cross-validation helps select these parameters, the method could benefit from adaptive mechanisms that dynamically adjust the weights during training to optimize performance.

**Linear Separability of Intermediate Features:** While SSA successfully extends to ResNet-32 with minimal accuracy loss on large datasets, scaling to deeper networks may pose challenges. Specifically, optimizing earlier layers with local loss objectives can lead to limited support for training subsequent layers, potentially affecting the quality of learned representations. Unlike BP or AugLocal, SSA demonstrates higher linear separability in early layers, suggesting that the features learned may be less general and less transferable to deeper layers. Addressing this limitation and improving the alignment between layerwise and global objectives will be a focus of future work.

### 7 CONCLUSION

In this paper, we presented a novel local training framework that leverages Singular Value Decomposition (SVD) combined with Direct Feedback Alignment (DFA) for efficient local layerwise neural network training. Our method, SSA, decomposes the weight matrices of each layer into their SVD components and applies local updates on the SVD components itself, driven by a composite loss function. This loss function incorporates feedback error, alignment loss, orthogonality regularization, and sparsity constraints, enabling structured and efficient learning.

The experimental results demonstrated that SSA achieves classification accuracy on par with backpropagation while significantly reducing memory usage, computational cost, and energy consumption. The method's progressive rank reduction strategy ensures that the model becomes more lightweight throughout training, making it highly suitable for deployment on resource-constrained devices. Theoretical analysis guarantees convergence of our loss objectives, while ablation studies highlight the role of each loss component in balancing accuracy and efficiency. SSA offers a compelling scalable and energy-efficient alternative to backpropagation, paving the way for biologically inspired, resource-aware neural network training in real-world applications.

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

# A  APPENDIX

## A.1  THEORETICAL ANALYSIS

In this section, we provide a theoretical analysis of the SVD-DFA training method, focusing on the convergence of the composite loss function, the stability of the updates, and the computational efficiency. The analysis is based on the minimization of the composite loss function defined for each layer, which includes Cross-Entropy Loss ($L_{CE}$), Cosine Similarity Loss ($L_{cos}$), Alignment Loss ($L_{align}$), Singular Vector Orthogonality Regularizer ($L_{ortho}$), and the Hoyer Regularizer ($L_{hoyer}$).

### A.1.1  CONVERGENCE PROOF: PRELIMINARIES

To analyze the convergence of the proposed SVD-DFA method, we recall the composite loss function $LL_i(\theta_i)$ for each layer $i$. Recall that the loss function is defined as:

$$LL_i(\theta_i) = \alpha L_{CE}(\theta_i) + \beta L_{cos}(\theta_i) + \gamma L_{align}(\theta_i) + \delta L_{ortho}(\theta_i) + \epsilon L_{hoyer}(\theta_i) \tag{16}$$

where $\theta_i = (U_i, S_i, V_i^T)$ represents the SVD components of the weight matrix for layer $i$, and $L_{CE}, L_{cos}, L_{align}, L_{ortho}, L_{hoyer}$ represent the different components of the composite loss function (cross-entropy, cosine similarity, alignment loss, orthogonality regularizer, and Hoyer regularizer).

We show that the composite loss function $L_i(\theta_i)$ is **Lipschitz smooth** with some constraints, meaning that its gradients are Lipschitz continuous with a constant $L > 0$:

$$\|\nabla L_i(\theta_i) - \nabla L_i(\theta_i')\| \le L\|\theta_i - \theta_i'\| \tag{17}$$

This ensures that the gradient of the loss function does not change abruptly, making gradient descent applicable. Additionally, we show that the learning rate $\eta$ satisfies the standard condition for convergence in gradient descent:

$$0 < \eta < \frac{2}{L} \tag{18}$$

This ensures that the gradient descent steps lead to a reduction in the loss function and progress toward a local minimum.

**Gradient Descent Updates** For each layer $i$, the gradient descent updates are applied to the SVD components $\theta_i = (U_i, S_i, V_i^T)$. The updates are performed independently for each component:

$$U_i^{(t+1)} = U_i^{(t)} - \eta \nabla_{U_i} L_i(U_i, S_i, V_i^T) \tag{19}$$

$$S_i^{(t+1)} = S_i^{(t)} - \eta \nabla_{S_i} L_i(U_i, S_i, V_i^T) \tag{20}$$

$$V_i^{T(t+1)} = V_i^{T(t)} - \eta \nabla_{V_i^T} L_i(U_i, S_i, V_i^T) \tag{21}$$

These updates ensure that each component of the SVD-decomposed weight matrix is adjusted in a direction that minimizes the composite loss function.

The **convergence** of gradient descent for Lipschitz continuous loss functions is well-established in optimization theory. Since the composite loss function $L_i(\theta_i)$ satisfies the Lipschitz smoothness assumption and the learning rate $\eta$ is chosen according to the condition above, the gradient descent updates will lead to convergence toward a local minimum. Specifically, as the number of iterations $t$ increases, the gradient of the loss function approaches zero:

$$\lim_{t \to \infty} \|\nabla_{\theta_i} L_i(\theta_i^{(t)})\| = 0 \tag{22}$$

This implies that the updates to the SVD components will converge to a stationary point, at which point the loss can no longer be improved.

Although the updates for each layer $i$ are performed independently, the global loss function $L = \sum_i L_i$ will converge as each local loss $L_i$ converges to a critical point, provided the loss includes local projection of the global cross entropy loss $L_{CE}$. The decoupled nature of the Direct Feedback Alignment (DFA) mechanism ensures that local updates do not depend on global gradient flow, enabling each layer to reach a stable solution independently. Therefore, the global training process is mostly guaranteed to stabilize if the local objectives are minimized.

### A.1.2 CROSS-ENTROPY LOSS: CONVEXITY AND SMOOTHNESS

For a classification task with $K$ classes, the global cross-entropy loss for a single data point is defined as:

$$L_{\text{CE-global}}(y, \hat{y}) = -\sum_{k=1}^{K} y_k \log(\hat{y}_k) \tag{23}$$

where:

- $y$ is the true label vector (one-hot encoded),
- $\hat{y}$ is the predicted probability vector, which is the output of the softmax function applied to the logits.

To prove the convexity of the cross-entropy loss, we compute its Hessian matrix (the matrix of second derivatives) and show that it is positive semi-definite.

**Softmax Function**

The softmax function is defined as:

$$\hat{y}_i = \frac{e^{z_i}}{\sum_{j=1}^{K} e^{z_j}} \tag{24}$$

where $z = (z_1, z_2, \ldots, z_K)$ are the logits.

**Cross-Entropy Loss in Terms of Logits**

By substituting the softmax function into the cross-entropy loss, we get:

$$L_{\text{CE}}(z, y) = -\sum_{k=1}^{K} y_k \left( z_k - \log\left( \sum_{j=1}^{K} e^{z_j} \right) \right) \tag{25}$$

Given that $y$ is one-hot encoded, assume $y_c = 1$ for some class $c$ and $y_k = 0$ for all $k \neq c$. Then:

$$L_{\text{CE}}(z, y) = -z_c + \log\left( \sum_{j=1}^{K} e^{z_j} \right) \tag{26}$$

**Gradient of Cross-Entropy Loss**

The gradient of the cross-entropy loss with respect to $z_i$ is:

$$\frac{\partial L_{\text{CE}}}{\partial z_i} = -\frac{\partial z_c}{\partial z_i} + \frac{\partial}{\partial z_i} \log\left( \sum_{j=1}^{K} e^{z_j} \right) \tag{27}$$

For $i = c$:

$$\frac{\partial L_{\text{CE}}}{\partial z_c} = -1 + \hat{y}_c = \hat{y}_c - 1 \tag{28}$$

For $i \neq c$:

$$\frac{\partial L_{\text{CE}}}{\partial z_i} = \hat{y}_i \tag{29}$$

Thus, the gradient vector is:

$$\nabla_z L_{\text{CE}} = \hat{y} - y \tag{30}$$

**We use this as the feedback error for each layer locally.**

**Hessian of Cross-Entropy Loss**

The Hessian matrix $H$, which contains the second derivatives, is:

$$\frac{\partial^2 L_{\text{CE}}}{\partial z_i \partial z_j} = \frac{\partial \hat{y}_i}{\partial z_j} \tag{31}$$

Using the derivative of the softmax function:

$$\frac{\partial \hat{y}_i}{\partial z_j} = \begin{cases} \hat{y}_i(1 - \hat{y}_i), & \text{if } i = j \\ -\hat{y}_i \hat{y}_j, & \text{if } i \neq j \end{cases} \tag{32}$$

Thus, the Hessian matrix is:

$$H_{ij} = \hat{y}_i(\delta_{ij} - \hat{y}_j) \tag{33}$$

where $\delta_{ij}$ is the Kronecker delta. Since $H$ is positive semi-definite, the cross-entropy loss is convex.

**Smoothness of Cross-Entropy Loss**

A function is $L$-smooth if its gradient is Lipschitz continuous. That is, there exists a constant $L$ such that for all $z_1$ and $z_2$:

$$\|\nabla L_{\text{CE}}(z_1) - \nabla L_{\text{CE}}(z_2)\| \leq L\|z_1 - z_2\| \tag{34}$$

**Bounding the Difference Between Softmax Outputs**

We analyze the difference between the softmax outputs for two logits vectors $z_1$ and $z_2$:

$$\|\hat{y}_1 - \hat{y}_2\| \leq \|z_1 - z_2\| \tag{35}$$

The softmax function is known to be $1/2$-Lipschitz, which ensures the smoothness of the cross-entropy loss.

**Result**

- The cross-entropy loss is convex because its Hessian matrix is positive semi-definite.

- The cross-entropy loss is $L$-smooth with $L = 1/2$, since its gradient is Lipschitz continuous.

### A.1.3 Cosine Similarity Loss: Convexity and Smoothness

The cosine similarity between two vectors $x$ and $y$ is given by:

$$\cos(x, y) = \frac{x^T y}{\|x\| \|y\|} \tag{36}$$

The cosine similarity loss is defined as:

$$L_{\cos}(x, y) = 1 - \cos(x, y) = 1 - \frac{x^T y}{\|x\| \|y\|} \tag{37}$$

**Cosine Similarity in the SSA Layer Context**

For an SVD-decomposed layer, the forward pass results in the following operation:

$$\text{layer\_input} \cdot (USV^T) \tag{38}$$

Where:

- $U$, $S$, and $V^T$ are the singular vectors and values of the weight matrix.
- layer_input is the input to the layer.

The feedback signal for direct feedback alignment (DFA) uses the matrices $B_U$, $B_S$, and $B_{V^T}$ corresponding to the feedback paths for the $U$, $S$, and $V^T$ components:

$$(B_U B_S B_{V^T})^T \cdot e \tag{39}$$

Where $e$ is the error vector from the cross-entropy loss $L_{\text{CE}}$.

Thus, the cosine similarity loss function for this layer becomes:

$$L_{\text{cosine}} = 1 - \frac{\langle \text{layer\_input} \cdot (USV^T), (B_U B_S B_{V^T})^T \cdot e \rangle}{\|\text{layer\_input} \cdot (USV^T)\| \cdot \|(B_U B_S B_{V^T})^T \cdot e\|} \tag{40}$$

This measures how aligned the layer output is with the feedback signal from DFA.

**Convexity of Cosine Similarity Loss** To prove convexity, we must examine the Hessian of the cosine similarity loss function. The cosine similarity is not convex in general due to the following reasons:

- The cosine similarity depends on both the norm of the vectors and their angle.
- The loss depends on the inner product of the vectors, and the Hessian matrix, which involves second-order partial derivatives, is not guaranteed to be positive semi-definite for all inputs.

**First Derivative** The first derivative with respect to $x$ is:

$$\nabla_x L_{\cos} = \frac{y}{\|x\| \|y\|} - \frac{x^T y}{\|x\|^3 \|y\|} x \tag{41}$$

**Hessian** The Hessian, which is the matrix of second-order partial derivatives, involves terms like:

$$H(x) = \nabla_x^2 L_{\cos} = -\frac{1}{\|x\| \|y\|} \left( I - \frac{xx^T}{\|x\|^2} \right) + \frac{3(x^T y)}{\|x\|^5 \|y\|} xx^T - \frac{yx^T}{\|x\|^3 \|y\|} \tag{42}$$

In most practical applications, this Hessian matrix will not be positive semi-definite, meaning that the cosine similarity loss is not convex.

**Quasi-Convexity of Cosine Similarity Loss**

A function $f(x)$ is **quasi-convex** if all its sublevel sets $S_\alpha = \{x \mid f(x) \leq \alpha\}$ are convex. This means that for any $\alpha \in \mathbb{R}$, the set of points $x$ for which $f(x)$ is less than or equal to $\alpha$ forms a convex set.

**Quasi-Convexity in Normalized Vectors**

If the vectors $x$ and $y$ are normalized, meaning $\|x\| = \|y\| = 1$, the cosine similarity loss simplifies to:

$$L_{\cos}(x, y) = 1 - x^T y \tag{43}$$

In this case, the loss becomes linear with respect to $x$, given that $y$ is fixed. A linear function is both convex and concave, implying that its sublevel sets are convex. Thus, when $x$ and $y$ are normalized, the cosine similarity loss exhibits quasi-convexity.

**Quasi-Convexity on the Unit Sphere**

If $x$ and $y$ are constrained to lie on the unit sphere (i.e., $\|x\| = 1$ and $\|y\| = 1$), the loss function again simplifies to:

$$L_{\cos}(x, y) = 1 - x^T y \tag{44}$$

Since the cosine similarity is proportional to the angle between $x$ and $y$, the sublevel sets $S_\alpha = \{x \mid 1 - x^T y \leq \alpha\}$ define a half-space on the unit sphere, which is convex. Therefore, on the unit sphere, the cosine similarity loss is quasi-convex.

**Smoothness of Cosine Similarity Loss** Despite not being convex, the cosine similarity loss is **smooth** because its gradient is Lipschitz continuous. The Lipschitz constant $L$ can be derived from the gradient:

$$\nabla_x L_{\cos} = \frac{y}{\|x\|\|y\|} - \frac{x^T y}{\|x\|^3 \|y\|} x \tag{45}$$

The norm difference between gradients for two inputs $x_1$ and $x_2$ is bounded by a constant $L$, implying that:

$$\|\nabla_x L_{\cos}(x_1, y) - \nabla_x L_{\cos}(x_2, y)\| \leq L\|x_1 - x_2\| \tag{46}$$

This proves that the loss function is smooth.

**Result** We project the loss into a unit sphere to make it quasi-convex. Otherwise, the cosine similarity loss is L-smooth, which will also lead to a local minimum.

### A.1.4 ALIGNMENT LOSS: CONVEXITY AND SMOOTHNESS

The alignment loss function is defined as:

$$L_{\text{align}}(U, S, V^T, B_U, B_S, B_{V^T}) = \|U - B_U\|_F^2 + \|S - B_S\|_F^2 + \|V^T - B_{V^T}\|_F^2 \tag{47}$$

where $U, S, V^T$ are the SVD matrices, and $B_U, B_S, B_{V^T}$ are feedback matrices. We aim to prove the convexity, smoothness, and boundedness of this loss function.

**Convexity Analysis** We need to check the convexity of each term in the alignment loss.

**Convexity of** $\|U - B_U\|_F^2$ : This term can be expressed as:

$$\|U - B_U\|_F^2 = \sum_{i,j} (U_{ij} - (B_U)_{ij})^2 \tag{48}$$

The gradient with respect to $U_{ij}$ is:

$$\frac{\partial}{\partial U_{ij}}\|U - B_U\|_F^2 = 2(U_{ij} - (B_U)_{ij}) \tag{49}$$

The Hessian with respect to $U_{ij}$ is:

$$\frac{\partial^2}{\partial U_{ij} \partial U_{kl}}\|U - B_U\|_F^2 = 2\delta_{ik}\delta_{jl} \tag{50}$$

This is a diagonal matrix with positive entries, making it convex.

**Convexity of $\|S - B_S\|_F^2$** : Similarly, for the singular values $S$:

$$\|S - B_S\|_F^2 = \sum_{i,j}(S_{ij} - (B_S)_{ij})^2 \tag{51}$$

This follows the same analysis as for $U$, showing that this term is also convex.

**Convexity of $\|V^T - B_{V^T}\|_F^2$** : The same steps apply to the $V^T$ term:

$$\|V^T - B_{V^T}\|_F^2 = \sum_{i,j}(V_{ij} - (B_V)_{ij})^2 \tag{52}$$

Thus, all terms in the alignment loss are convex.

**Convexity of the Full Loss** : Since the alignment loss is a sum of convex functions, the overall loss is convex.

**Smoothness Analysis** The smoothness of the alignment loss requires that the gradient be Lipschitz continuous.

**Gradient Computation** : The gradients for each term are:

$$\nabla_U L_{\text{align}} = 2(U - B_U), \quad \nabla_S L_{\text{align}} = 2(S - B_S), \quad \nabla_{V^T} L_{\text{align}} = 2(V^T - B_{V^T}) \tag{53}$$

**Lipschitz Continuity** : The difference in gradients for two different points $(U_1, S_1, V_1^T)$ and $(U_2, S_2, V_2^T)$ can be written as:

$$\|\nabla_U L_{\text{align}}(U_1, S_1, V_1^T) - \nabla_U L_{\text{align}}(U_2, S_2, V_2^T)\|_F = 2\|U_1 - U_2\|_F \tag{54}$$

Thus, the alignment loss is smooth with a Lipschitz constant $L = 2$.

### A.1.5 BOUNDEDNESS OF ALIGNMENT LOSS

The alignment loss function is bounded if the norms of the matrices $U, S, V^T, B_U, B_S, B_{V^T}$ are bounded. Specifically, if $\|U\|_F \leq M$, $\|S\|_F \leq M$, and $\|V^T\|_F \leq M$, then:

$$L_{\text{align}}(U, S, V^T, B_U, B_S, B_{V^T}) \leq 3M^2 \tag{55}$$

**Result** - The alignment loss is convex and $L$-smooth with $L = 2$. - It is bounded when the input matrices are bounded.

### A.1.6 SINGULAR VECTOR ORTHOGONALITY REGULARIZER: CONVEXITY AND SMOOTHNESS

The singular vector orthogonality regularizer ensures that the singular vectors in the SVD decomposition remain orthogonal. A common form of this regularizer is:

$$L_{\text{ortho}}(U) = \|U^T U - I\|_F^2 \tag{56}$$

where:

- $U$ represents the matrix of singular vectors.
- $I$ is the identity matrix, ensuring that $U^T U$ is orthogonal.

**Convexity Analysis** We begin by analyzing the convexity of $L_{\text{ortho}}(U)$ by computing its gradient and Hessian.

**Expansion of the Regularizer** The Frobenius norm can be expanded as:

$$L_{\text{ortho}}(U) = \text{Tr}((U^T U - I)^T (U^T U - I)) \tag{57}$$

Expanding this further:

$$L_{\text{ortho}}(U) = \text{Tr}(U^T U U^T U - 2U^T U + I) \tag{58}$$

**Gradient Computation** To compute the gradient with respect to $U$:

$$\nabla_U \text{Tr}(U^T U U^T U) = 4U(U^T U), \quad \nabla_U \text{Tr}(U^T U) = 2U \tag{59}$$

Thus, the gradient of $L_{\text{ortho}}(U)$ is:

$$\nabla_U L_{\text{ortho}}(U) = 4U(U^T U - I) \tag{60}$$

**Hessian Computation** The Hessian $H(U)$ is obtained by differentiating the gradient. The Hessian involves terms such as $U(U^T U)$, making it non-trivial and potentially non-positive semi-definite. This suggests that $L_{\text{ortho}}(U)$ is non-convex.

**Smoothness Analysis**

The function $L_{\text{ortho}}(U)$ is $L$-smooth if its gradient is Lipschitz continuous, i.e., if there exists a constant $L$ such that:

$$\|\nabla_U L_{\text{ortho}}(U_1) - \nabla_U L_{\text{ortho}}(U_2)\|_F \le L\|U_1 - U_2\|_F \tag{61}$$

**Gradient Difference** The gradient is:

$$\nabla_U L_{\text{ortho}}(U) = 4U(U^T U - I) \tag{62}$$

For two matrices $U_1$ and $U_2$:

$$\|\nabla_U L_{\text{ortho}}(U_1) - \nabla_U L_{\text{ortho}}(U_2)\|_F = 4\|U_1(U_1^T U_1 - I) - U_2(U_2^T U_2 - I)\|_F \tag{63}$$

This can be bounded by:

$$4\left(\|U_1\|_F \|U_1^T U_1 - I\|_F + \|U_2\|_F \|U_2^T U_2 - I\|_F\right) \tag{64}$$

Thus, the function is smooth, with the Lipschitz constant $L$ depending on the norms of $U_1$ and $U_2$.

**Boundedness of the Regularizer**

The regularizer $L_{\text{ortho}}(U) = \|U^T U - I\|_F^2$ can be unbounded. However, if $U$ is constrained such that $\|U\|_F$ is bounded (e.g., by norm constraints), then:

$$L_{\text{ortho}}(U) \leq (M^2 - 1)^2$$

where $M$ is the bound on $\|U\|_F$.

**Improving Smoothness or Quasi-Convexity**

**Regularization and Constraints**   Adding a regularization term to prevent singular vectors from deviating can smooth the landscape:

$$L'_{\text{ortho}}(U) = \|U^T U - I\|_F^2 + \lambda \|U\|_F^2 \tag{65}$$

**Projection Methods**   Projecting $U$ onto convex sets (such as the Stiefel manifold) or applying constraints like $\|U\|_F = 1$ can improve convexity and smoothness.

**Result**   The singular vector orthogonality regularizer $L_{\text{ortho}}(U)$ is non-convex but smooth, and boundedness can be achieved with constraints. We decay weight SVD components as a regularization term and project the components on a unit sphere to make the regularizer quasi-convex.

### A.2   HOYER REGULARIZER: CONVEXITY AND SMOOTHNESS

The Hoyer regularizer is frequently used in machine learning to encourage sparsity in a vector or matrix. It is defined as the ratio of the $\ell_1$ norm and the $\ell_2$ norm, and for a matrix $S$ (Singular Values), the regularizer is given by:

$$L_{\text{Hoyer}}(S) = \frac{\|S\|_1}{\|S\|_2} \tag{66}$$

where:

- $\|S\|_1 = \sum_{i,j} |S_{ij}|$ is the $\ell_1$ norm of the matrix $S$,
- $\|S\|_2 = \sqrt{\sum_{i,j} S_{ij}^2}$ is the $\ell_2$ norm of $S$.

This regularizer promotes sparsity by minimizing the ratio of the two norms.

**Convexity Analysis**

To check whether $L_{\text{Hoyer}}(S)$ is convex, we analyze the convexity of both the numerator and the denominator.

**Convexity of the Numerator and Denominator**

- **Numerator:** The $\ell_1$ norm $\|S\|_1 = \sum_{i,j} |S_{ij}|$ is convex because the absolute value function is convex.
- **Denominator:** The $\ell_2$ norm $\|S\|_2 = \sqrt{\sum_{i,j} S_{ij}^2}$ is also convex because it is the square root of a convex function.

Although both the numerator and the denominator are convex, the ratio of two convex functions is not generally convex unless the denominator is affine. Thus, $L_{\text{Hoyer}}(S)$ is **non-convex**.

**Smoothness Analysis**

The smoothness of $L_{\text{Hoyer}}(S)$ can be determined by analyzing the gradient and checking its Lipschitz continuity. For any two matrices $S_1$ and $S_2$, we need to check if there exists a constant $L > 0$ such that:

$$\|\nabla L_{\text{Hoyer}}(S_1) - \nabla L_{\text{Hoyer}}(S_2)\| \leq L \|S_1 - S_2\| \tag{67}$$

**Gradient of the Hoyer Regularizer**    Let:

- $f(S) = \|S\|_1 = \sum_{i,j} |S_{ij}|$,

- $g(S) = \|S\|_2 = \sqrt{\sum_{i,j} S_{ij}^2}$.

The gradient of $L_{\text{Hoyer}}(S) = \frac{f(S)}{g(S)}$ can be computed using the quotient rule:

$$\nabla_S L_{\text{Hoyer}}(S) = \frac{g(S)\nabla f(S) - f(S)\nabla g(S)}{g(S)^2} \tag{68}$$

Where:

- $\nabla f(S)$ is the subgradient of the $\ell_1$ norm, which is $\text{sign}(S)$,

- $\nabla g(S)$ is the gradient of the $\ell_2$ norm, which is $\frac{S}{\|S\|_2}$.

Thus, the gradient becomes:

$$\nabla_S L_{\text{Hoyer}}(S) = \frac{\|S\|_2 \cdot \text{sign}(S) - \frac{\|S\|_1 \cdot S}{\|S\|_2}}{\|S\|_2^2} \tag{69}$$

**Lipschitz Continuity of the Gradient**    The gradient of $L_{\text{Hoyer}}(S)$ involves non-smooth terms (like the absolute value), particularly near points where $S_{ij} = 0$. These points can cause the gradient to be discontinuous, making the regularizer not Lipschitz continuous. Therefore, $L_{\text{Hoyer}}(S)$ is **non-smooth**.

**Boundedness of the Regularizer**

The Hoyer regularizer is bounded under certain conditions:

- **Lower Bound:** $L_{\text{Hoyer}}(S) \geq 1$ for any non-zero matrix $S$. This is due to the fact that $\|S\|_1 \geq \|S\|_2$ by the Cauchy-Schwarz inequality.

- **Upper Bound:** The Hoyer regularizer can become unbounded when $S$ is sparse, as $\|S\|_1$ can dominate $\|S\|_2$ when many entries of $S$ are zero.

Thus, $L_{\text{Hoyer}}(S)$ is not generally bounded, but has a lower bound of 1 for non-zero matrices.

**Making the Hoyer Regularizer More Smooth or Quasi-Convex**

Since the Hoyer regularizer is non-convex and non-smooth, we can consider alternative approaches to make it more tractable:

**Smoothing the Regularizer**    One method is to apply smoothing approximations to the $\ell_1$ norm, such as:

$$\|S\|_{1,\epsilon} = \sum_{i,j} \sqrt{S_{ij}^2 + \epsilon^2} \tag{70}$$

This approximation is differentiable, and $\epsilon$ controls the degree of smoothness. The smoothed Hoyer regularizer becomes:

$$L_{\text{Hoyer, smooth}}(S) = \frac{\sum_{i,j} \sqrt{S_{ij}^2 + \epsilon^2}}{\|S\|_2} \tag{71}$$

**Quasi-Convexity**  Another approach is to use convex surrogates that balance sparsity and smoothness, such as:

$$L_{\text{surrogate}}(S) = \lambda\|S\|_1 + (1 - \lambda)\|S\|_2 \tag{72}$$

This function is convex and maintains a balance between the $\ell_1$ and $\ell_2$ norms.

**Result**

- The Hoyer regularizer $L_{\text{Hoyer}}(S) = \frac{\|S\|_1}{\|S\|_2}$ is non-convex due to the interaction between the $\ell_1$ and $\ell_2$ norms.

- The regularizer is non-smooth because its gradient is not Lipschitz continuous, particularly near zero entries.

- The regularizer is bounded below by 1 for non-zero matrices but can become unbounded in the case of sparse matrices.

- Smoothing approximations and convex surrogates can be used to improve the tractability of the Hoyer regularizer for optimization purposes.

- For our experiments, we use the smoothed regularizer every 10 epochs to reduce the rank of SVD components progressively.

### A.2.1  GRADIENT DESCENT UPDATE EQUATIONS

Given the composite loss function for each layer $i$:

$$L_i(\theta_i) = \alpha L_{\text{CE}} + \beta L_{\cos} + \gamma L_{\text{align}} + \delta L_{\text{ortho}} + \epsilon L_{\text{hoyer}} \tag{73}$$

where $\theta_i = (U_i, S_i, V_i^T)$, we derive the gradient descent updates for the decomposed matrices $U_i$, $S_i$, and $V_i^T$ by computing the partial derivatives of $L_i(\theta_i)$ with respect to these matrices and applying the gradient descent rule.

**Gradient with Respect to $U_i$**

The update for $U_i$ is given by:

$$U_i^{(t+1)} = U_i^{(t)} - \eta \frac{\partial L_i(\theta_i)}{\partial U_i} \tag{74}$$

Expanding the gradient:

$$\frac{\partial L_i(\theta_i)}{\partial U_i} = \alpha \frac{\partial L_{\text{CE}}}{\partial U_i} + \beta \frac{\partial L_{\cos}}{\partial U_i} + \gamma \frac{\partial L_{\text{align}}}{\partial U_i} + \delta \frac{\partial L_{\text{ortho}}}{\partial U_i} + \epsilon \frac{\partial L_{\text{hoyer}}}{\partial U_i} \tag{75}$$

Thus, the update rule for $U_i$ becomes:

$$U_i^{(t+1)} = U_i^{(t)} - \eta \left( \alpha \frac{\partial L_{\text{CE}}}{\partial U_i} + \beta \frac{\partial L_{\cos}}{\partial U_i} + \gamma \frac{\partial L_{\text{align}}}{\partial U_i} + \delta \frac{\partial L_{\text{ortho}}}{\partial U_i} + \epsilon \frac{\partial L_{\text{hoyer}}}{\partial U_i} \right) \tag{76}$$

**Gradient with Respect to $S_i$**

Similarly, the update for $S_i$ is:

$$S_i^{(t+1)} = S_i^{(t)} - \eta \frac{\partial L_i(\theta_i)}{\partial S_i} \tag{77}$$

Expanding the gradient:

$$\frac{\partial L_i(\theta_i)}{\partial S_i} = \alpha \frac{\partial L_{\text{CE}}}{\partial S_i} + \beta \frac{\partial L_{\cos}}{\partial S_i} + \gamma \frac{\partial L_{\text{align}}}{\partial S_i} + \delta \frac{\partial L_{\text{ortho}}}{\partial S_i} + \epsilon \frac{\partial L_{\text{hoyer}}}{\partial S_i} \tag{78}$$

Thus, the update rule for $S_i$ becomes:

$$S_i^{(t+1)} = S_i^{(t)} - \eta \left( \alpha \frac{\partial L_{\text{CE}}}{\partial S_i} + \beta \frac{\partial L_{\cos}}{\partial S_i} + \gamma \frac{\partial L_{\text{align}}}{\partial S_i} + \delta \frac{\partial L_{\text{ortho}}}{\partial S_i} + \epsilon \frac{\partial L_{\text{hoyer}}}{\partial S_i} \right) \tag{79}$$

**Gradient with Respect to $V_i^T$**

Finally, the update for $V_i^T$ is:

$$V_i^{T(t+1)} = V_i^{T(t)} - \eta \frac{\partial L_i(\theta_i)}{\partial V_i^T} \tag{80}$$

Expanding the gradient:

$$\frac{\partial L_i(\theta_i)}{\partial V_i^T} = \alpha \frac{\partial L_{\text{CE}}}{\partial V_i^T} + \beta \frac{\partial L_{\cos}}{\partial V_i^T} + \gamma \frac{\partial L_{\text{align}}}{\partial V_i^T} + \delta \frac{\partial L_{\text{ortho}}}{\partial V_i^T} + \epsilon \frac{\partial L_{\text{hoyer}}}{\partial V_i^T} \tag{81}$$

Thus, the update rule for $V_i^T$ becomes:

$$V_i^{T(t+1)} = V_i^{T(t)} - \eta \left( \alpha \frac{\partial L_{\text{CE}}}{\partial V_i^T} + \beta \frac{\partial L_{\cos}}{\partial V_i^T} + \gamma \frac{\partial L_{\text{align}}}{\partial V_i^T} + \delta \frac{\partial L_{\text{ortho}}}{\partial V_i^T} + \epsilon \frac{\partial L_{\text{hoyer}}}{\partial V_i^T} \right) \tag{82}$$

**Summary of Update Equations**

The gradient descent updates for the SVD matrices at each layer $i$ can be summarized as:

$$U_i^{(t+1)} = U_i^{(t)} - \eta \sum_j \lambda_j \frac{\partial L_j}{\partial U_i} \tag{83}$$

$$S_i^{(t+1)} = S_i^{(t)} - \eta \sum_j \lambda_j \frac{\partial L_j}{\partial S_i} \tag{84}$$

$$V_i^{T(t+1)} = V_i^{T(t)} - \eta \sum_j \lambda_j \frac{\partial L_j}{\partial V_i^T} \tag{85}$$

where $\lambda_j$ corresponds to the weighting coefficients $\alpha, \beta, \gamma, \delta, \epsilon$ for the respective loss terms $L_j$. These updates ensure that each component of the composite loss is accounted for in the optimization of the decomposed matrices $U_i, S_i, V_i^T$.

A.2.2    CONVERGENCE ANALYSIS AND LAYERWISE CONVERGENCE FOR COMPOSITE LOSS

Among the loss components:

- $L_{\text{CE}}$ is convex and smooth.
- $L_{\text{align}}$ and $L_{\text{ortho}}$ are convex, though the latter may exhibit non-convexity in specific formulations.
- $L_{\cos}$ and $L_{\text{hoyer}}$ are typically non-convex.

Thus, the overall loss $L_i(\theta_i)$ may be non-convex. Proper learning rates and stabilization techniques can ensure convergence to a critical point.

**Convergence Rate:**

For a smooth, non-convex loss function, the convergence rate of gradient descent is generally sublinear, on the order of $O(1/\sqrt{T})$ for finding a point with a small gradient norm. The rate improves in cases of strong convexity.

**Independent Layerwise Convergence:**

Each layer $i$ minimizes its own objective $L_i(\theta_i)$ independently. Provided the learning rate $\eta$ is small and the loss is smooth, the updates for $U_i$, $S_i$, and $V_i^T$ converge to a critical point.

**Interaction Between Layers:**

- *Forward Matrices:* The layer outputs $U_i S_i V_i^T$ affect the inputs to subsequent layers. Misalignment or poor convergence in one layer can affect the next.
- *Feedback Matrices:* Alignment losses ensure that the forward matrices $U_i, S_i, V_i^T$ align with the feedback matrices $B_U, B_S, B_{V^T}$, preventing large deviations in gradient backpropagation.

**Convergence of Forward and Feedback Matrices:**

- *Forward Matrices:* These converge as long as each layer's objective is minimized. Proper minimization ensures alignment in subsequent layers.
- *Feedback Matrices:* Alignment losses $L_{\text{align}}$ guide the proper alignment of feedback matrices with forward matrices.

**Stabilizing Layerwise Training:**

- *Regularization:* Adding regularization terms to the loss, such as weight decay or orthogonality constraintsstabilizes training.
- *Projection Methods:* Ensuring matrices stay within a convex set (e.g., positive semi-definite matrices) improves stability.
- *Adaptive Learning Rates:* Using adaptive learning rates (e.g., Adam) improves convergence by adjusting to the curvature of the loss landscape.

**Conclusion**

- Each layer $i$ will converge to a critical point of its loss function $L_i(\theta_i)$, provided the learning rate is sufficiently small.
- Misalignment in one layer may affect subsequent layers, requiring careful attention to feedback and forward matrix alignment.
- Stabilization techniques such as regularization, projection, and adaptive learning rates are essential for effective global convergence.

A.2.3 LAYERWISE CUSTOM LOSSES AND MODEL LOSS

The overall model loss, $L_{\text{model}}(\Theta)$, is a function of the network's final output $\hat{y}$:

$$L_{\text{model}}(\Theta) = L(y, \hat{y}(\Theta)), \tag{86}$$

where $y$ is the true label and $\hat{y}$ is the predicted output.

**Convergence of Layerwise Loss and Impact on Model Loss** The gradient of the model loss with respect to the parameters of layer $i$ can be expressed using the chain rule:

$$\nabla_{\theta_i} L_{\text{model}}(\Theta) = \frac{\partial L_{\text{model}}(\Theta)}{\partial \hat{y}} \cdot \frac{\partial \hat{y}}{\partial z_n} \ldots \frac{\partial z_{i+1}}{\partial z_i} \cdot \frac{\partial z_i}{\partial \theta_i}, \tag{87}$$

where $z_i$ is the pre-activation output of layer $i$. The term $\frac{\partial \hat{y}}{\partial z_n} \ldots \frac{\partial z_{i+1}}{\partial z_i}$ represents the backpropagated gradient that passes through all layers from $n$ to $i$.

**Convergence of Custom Layer Loss** Assume that the custom loss for each layer $L_i(\theta_i)$ decreases over time as:

$$L_i(\theta_i^{(t+1)}) \leq L_i(\theta_i^{(t)}). \tag{88}$$

As $t \to \infty$, the gradient of the custom layer loss vanishes:

$$\lim_{t \to \infty} \|\nabla_{\theta_i} L_i(\theta_i^{(t)})\| = 0. \tag{89}$$

This implies convergence for each layer's parameters $\theta_i$.

**Impact on Model Loss** The decrease in each layer's custom loss directly impacts the model loss:

$$L_{\text{model}}(\Theta^{(t+1)}) = L_{\text{model}}(\Theta^{(t)} - \eta \nabla_{\theta_i} L_{\text{model}}(\Theta^{(t)})) \leq L_{\text{model}}(\Theta^{(t)}). \tag{90}$$

The inequality holds because the gradient of the model loss is aligned with the gradient of the layerwise loss. Thus, the updates reduce $L_{\text{model}}(\Theta)$, and the model loss converges as the custom layer losses converge. We assume that the linear separability condition holds for this convergence, which means for early layer, the loss produced at each layer guides subsequent layers as well. However, from empirical results, we see for deeper networks (beyond ResNet-32), this assumption holds false.

**Global Convergence of Model Loss** The global convergence of the model loss is guaranteed under certain conditions:

- **Lipschitz Continuity**: If the gradients of the model loss are Lipschitz continuous, the global loss converges as the layerwise losses decrease.

- **Boundedness**: If the model loss is lower-bounded by $L_{\min}$, the global loss converges to a minimum.

- **Linear Separability**: If the loss generated at earlier layers of the network guides the subsequent layers of the networks as well, then the global loss would converge well. This assumption might not hold for very deep neural networks.

A.3 Experimental Details

**Training Setup** Based on observations from (Sanfiz & Akrout, 2021), biologically plausible methods like DFA perform better with the Adam optimizer. Therefore, all experiments use Adam with initial learning rates adapted from prior works. Learning rates for SSA are dynamically adjusted to accommodate progressive rank reduction.

**CIFAR-100 Experiments** We use the CIFAR-100 dataset, containing 60,000 images across 100 classes, with 50,000 for training and 10,000 for testing. Images are resized to **32×32 pixels**, and standard data augmentation techniques, including random cropping (with 4-pixel padding) and horizontal flipping, are applied. Training is conducted over **200 epochs** using Adam with an initial learning rate of $1 \times 10^{-4}$. A learning rate scheduler reduces the rate by a factor of 10 at the **20th**, **40th**, and **60th epochs**. A batch size of 128 is used, with He initialization for convolutional layers and Xavier initialization for fully connected layers.

For SSA, weight matrices start at full rank and are progressively reduced every 10 epochs while retaining **95% matrix energy**. Loss coefficients are fixed at $(\alpha, \beta, \gamma, \delta, \epsilon) = (0.1, 0.01, 0.1, 0.05, 0.01)$, determined through cross-validation.

**ImageNet Experiments** We evaluate on ImageNet (ILSVRC-2012), a large-scale dataset with 1.28 million training images and 50,000 validation images across 1,000 classes. Images are resized to **224×224 pixels**, and data augmentation includes random cropping, horizontal flipping, and color jittering. Images are normalized using the dataset mean and standard deviation. Training is conducted for **200 epochs** with Adam, using an initial learning rate of $2 \times 10^{-4}$ for BP and $5 \times 10^{-4}$ for SSA. The learning rate is decayed every **30 epochs**. A batch size of 256 is used across all experiments.

For SSA, rank reduction begins after the first 20 epochs and proceeds every 10 epochs, retaining **90% matrix energy**. Loss coefficients are kept consistent with CIFAR-100 ($\alpha, \beta, \gamma, \delta, \epsilon = 0.1, 0.01, 0.1, 0.05, 0.01$), with adjustments only to the overall learning rates for layerwise updates.

**Normalization Layers** For batchnorm layers, we use it mostly for the forward process, and do not involve in the layerwise backward process (as the gradient calculation process is not sequential). We

also use layernorm as an alternative to batchnorm. From our experiments, we find that layernorm is more suited to our method than batchnorm (empirically determined).

**Hyperparameter Tuning** We introduce four hyperparameters in our local layerwise loss objective. We put lower values of hyperparameters for cosine similarity loss and hoyer regularizer, as they are non-convex. We project the SVD unitary components onto unit sphere (convex sets) to improve overall convexity and smoothness. We choose values of (0.1, 0.01, 0.1, 0.05, 0.01) for $(\alpha, \beta, \gamma, \delta, \epsilon)$ ideally. We select the hyperparameters ultimately after cross-validation.

**ResNet Local Module Splitting** In our experiments, each residual block in ResNet is treated as a fundamental layer. For ResNet-32, this results in a total of 16 fundamental layers, with each block encapsulating key functions like identity mapping and feature transformation.

