# OpenReview forum: "Unlocking SVD-Space for Feedback Aligned Local Training"
_ICLR.cc/2025/Conference — Submitted to ICLR 2025_

### Official Review · Reviewer_dZya · 2024-10-28

**Soundness:** 2
**Presentation:** 2
**Contribution:** 3
**Rating:** 3
**Confidence:** 4

**Summary:**

The authors improve the Direct Feedback Alignment (DFA) algorithm to make it more efficient through low-rank decomposition. The weight and feedback matrices are decomposed using SVD, and local losses are used to encourage alignment and sparsity, and preserve orthogonality of the U and V matrices.

The algorithm is competitive with backpropagation on different models and image classification tasks, while providing attractive gains in terms of time and memory.

**Strengths:**

- Combining DFA with low rank factorization is a good idea.
- The proposed algorithm is relatively simple and intuitive.
- The results are competitive with backpropagation, with encouraging time and memory gains.
- The authors compare their method with an extensive spectrum of other alternatives to backpropagation.
- The ablation study is very relevant and clearly shows the impact of each loss on accuracy and computation time

**Weaknesses:**

1. As SSA is essentially a low-rank approximation of DFA, I would expect a more extensive comparison with DFA, as the proposed method is built upon it. As far as I know, DFA has successfully been applied to larger models as used in the paper (see Sanfiz et al. (2021), https://arxiv.org/pdf/2108.13446). I am unsure what “omitted from the next Table 2 due to their inability to scale to larger networks” means.

    Also, it would be expected that SSA at best matches DFA. On table 1 however it outperforms it by 13%. This is likely due to the alignment losses in SSA, and I believe that applying the same losses to DFA could provide a fairer baseline. It would also allow us to quantify the loss in accuracy caused by the low-rank factorization.

2. I could be beneficial to include models different than CNNs, and tasks different than image classification. For instance, a simple MLP.
3. The method introduces 4 to 5 new hyperparameters, which currently have to be tuned. It would be interesting to measure the impact of these on the final accuracy.
4. Presentation and formatting: The citations are not correctly inserted in the text, the parentheses are missing. Use the \citep command instead if using natbib. Less important, but I believe the tables and figure 1 could benefit from a cleaner layout / style to improve readability and visibility of the results.

    Eq. 6 is unclear to me. This quantity is the error (i.e. the gradient of the loss w.r.t the last hidden states), not the crossentropy loss. I find the notations very confusing. In addition, the layers do not receive the gradients of the cross-entropy loss but instead use DFA to approximate it, which is not reflected in equations 5 and 6.

5. The theoretical analysis in the Appendix is wrong in many ways. For instance:
    - At the beginning, on line 613, it is stated that the composite loss is Lipschitz smooth, which is clearly not the case given that the cosine similarity is not.
    - The loss functions are also dependent on the output of the previous layer and on the error, which is not taken into account here. This means that the local loss is not always decreasing, as updating layer 1 can impact the loss of layer 2 for example.
    - In A.1.6, the claimed “Lipschitz constant” of $\nabla_U L_\text{ortho}(U)$ depends on U, which is forbidden.

    I would strongly advise the authors to revise this part. It would also be best to make it more concise, for instance A.2.1 is trivial, and there is no need to take so much space proving a function is smooth, especially when it is not (see the part about cosine similarity).

    However this analysis does not impact the contributions of the paper, and the algorithm does not rely on it.

I believe that the idea is interesting and am willing to changing my score if the authors address my concerns.

**Questions:**

1. I do not understand how convolution are handled by SSA. How do you perform SVD decomposition on the convolution layers?
2. What is the motivation behind the cosine similarity loss? I do not get why aligning the output of the layer with its estimated gradient could lead to better learning (although I do see that this is the case from the ablation study).

---

> ### Author Response · Authors · 2024-11-23
> **Response to Reviewer pt 1**
>
> We appreciate the reviewer’s detailed feedback. We will address the queries below:
>
> i) As SSA is essentially a low-rank approximation of DFA, I would expect a more extensive comparison with DFA, as the proposed method is built upon it.
>
> Authors: We have added a detailed discussion of SSA versus DFA and its variants in updated Section 6.1 as present in Sanfiz et al. (2021), https://arxiv.org/pdf/2108.13446.
>
> ii) Also, it would be expected that SSA at best matches DFA. On table 1 however it outperforms it by 13%. This is likely due to the alignment losses in SSA, and I believe that applying the same losses to DFA could provide a fairer baseline. It would also allow us to quantify the loss in accuracy caused by the low-rank factorization.
>
> Authors: The custom loss operates on the SVD-components of the forward and feedback weights. We wouldn't be able to apply the same loss to normal weights. We begin our training on the full-rank of SVD-decomposed matrices itself, and then reduce the rank while training, applying different loss components to align the gradient direction and the weight-spaces between the feedback and forward signals. As we are training while reducing the rank, the loss due to rank-reduction gets adjusted in training.
>
> iii) The method introduces 4 to 5 new hyperparameters, which currently have to be tuned. It would be interesting to measure the impact of these on the final accuracy.
>
> Authors: We appreciate the reviewer’s concern about the impact of hyperparameters on the final accuracy. We have added a detailed explanation of our hyperparameter choices in Section 4 Experimental Setup. The loss-related hyperparameters (α,β,γ,δ,ϵ) were kept consistent across all experiments, with adjustments made only to the overall learning rate to account for differences between smaller datasets (e.g., CIFAR-10) and larger datasets (e.g., ImageNet).
>
> As layerwise loss objectives are largely independent, tuning involves optimizing simpler, localized objectives. This allowed us to perform k=3-fold cross-validation on a small grid of candidate values, taking only 5–10% of the total training time. We verified the robustness of the hyperparameters under these conditions.
>
> While this paper focuses on demonstrating the feasibility of SSA rather than exhaustive hyperparameter optimization, we acknowledge that further exploration of optimal tuning strategies is valuable and have noted this in Section 6.2 Limitations. We thank the reviewer for highlighting this area for potential future work.
>
> iv) Eq. 6 is unclear to me. This quantity is the error (i.e. the gradient of the loss w.r.t the last hidden states), not the crossentropy loss. I find the notations very confusing. In addition, the layers do not receive the gradients of the cross-entropy loss but instead use DFA to approximate it, which is not reflected in equations 5 and 6.
>
> Authors: We acknowledge the mistakes in the notations and have corrected them in the revised paper (Section 3.3). Additionally, we updated Fig. 1 to clarify the notations and illustrate the gradient input to the layers more effectively.
>
> The local loss comprises cross-entropy loss, alignment loss, cosine similarity loss, a singular vector orthogonality regularizer, and the Hoyer regularizer. The gradient of this composite loss is provided as input to the layers, with the feedback error (from DFA) specifically corresponding to the gradient of the cross-entropy loss. These updates aim to improve the clarity and understanding of our method.
>
> v) The theoretical analysis in the Appendix is wrong in many ways.
>
> Authors: We appreciate the reviewer’s observation about the theoretical analysis. While the cosine similarity component or orthogonal regularizer component is not inherently Lipschitz smooth, we address this by applying constraints that ensure bounded gradients. Specifically, we normalize the norms of
> x and
> y to 1, effectively projecting the non-convex cosine similarity loss or orthogonal loss onto a convex sublevel set. This approach bounds the gradient differences, making the loss components quasi-convex and Lipschitz smooth under these conditions. These constraints are integrated into our custom loss function to maintain theoretical consistency.
> We will revise the part where we say loss will be always decreasing globally, in our theroretical analysis as pointed out by the reviewer.

---

> > ### Author Response · Authors · 2024-11-23
> > **Response to Reviewer pt 2**
> >
> > vi) I do not understand how convolution are handled by SSA. How do you perform SVD decomposition on the convolution layers?
> >
> > Authors:  A detailed explanation of the SVD approach for convolutional layers can be found in Section 3.1 of the revised paper.
> >
> > To summarize, the convolutional kernel $K \in \mathbb{R}^{N \times C \times H \times W}$ is first reshaped into a 2D matrix $K' \in \mathbb{R}^{NW \times CH}$. Singular Value Decomposition (SVD) is then applied: $ K' = U \Sigma V^T, $ where $U$, $\Sigma$, and $V^T$ denote the singular vectors and singular values, with $r = \min(NW, CH)$. The resulting components are reshaped into two smaller convolutional kernels: $U \sqrt{\Sigma}$ is reshaped into $K_1 \in \mathbb{R}^{r \times C \times H \times 1}$, $\sqrt{\Sigma} V^T$ is reshaped into $K_2 \in \mathbb{R}^{N \times r \times 1 \times W}$.
> >
> > This approach splits the original convolution into two consecutive layers, which preserves both spatial and hierarchical features while reducing computational complexity. Gradients are computed directly for $K_1$ and $K_2$ during backpropagation, and the decomposed kernels are retained for inference. We believe this explanation addresses the reviewer’s concerns.
> >
> > vii) What is the motivation behind the cosine similarity loss? I do not get why aligning the output of the layer with its estimated gradient could lead to better learning (although I do see that this is the case from the ablation study).
> >
> > Authors: We have seen in DFA papers that the angle between the true gradient (from BP) and the DFA gradient must be aligned to ensure updates occur in the correct direction. In our method, since BP gradients (true gradients) are unavailable, we use the forward signal as a close approximation for the cosine similarity loss. This ensures that the gradient direction remains aligned, even without the exact BP gradients. To emphasize direction alignment without overwhelming other components, we assign a lower hyperparameter weight to this term.
> >
> > We hope these updates address the reviewer’s concerns and welcome any additional suggestions or clarifications. We thank the reviewer for their valuable feedback and for helping us improve the clarity and depth of our work.

---

> ### Comment · Reviewer_dZya · 2024-12-03
>
> I thank the authors for their answers, which addressed a few of my concerns. I also acknowledge that the updated paper has improved.
>
> ---
>
> I still see a lot of issues in the theoretical analysis of Appendix A.1. In particular I see that errors that I mentioned in my previous message have not been addressed.
> - The local losses are not independent: modifying the weights of layer $i$ will impact the input of a later layer $j$, as well as the error... Yes, if we were to optimize a single layer (with the other onesfrozen) then maybe it could converge, but if all layers are optimized at the same time there is no guarantee of convergence at all for any of the losses.
> - The orthogonal regularization is NOT Lipschitz smooth
> - There is no proof of convergence of DFA in the general case, so how can we get any theoretical insight of the convergence of the model after adding more losses?
>
> The authors go through long and sketchy proofs to try to prove the convexity and Lipschitz smoothness of the local losses, which is still wrong in its current form. Moreover they use these results to assess the convergence of the global loss. This is equivalent to proving the convergence of an arbitrary neural network, which is not possible in the general case given that it is not convex not Lipschitz smooth.
>
> **I am starting to think that the proof has been written by an LLM.**
>
> ---
>
> > The custom loss operates on the SVD-components of the forward and feedback weights. We wouldn't be able to apply the same loss to normal weights.
> >
>
> I beg to disagree, the cosine similarity loss does not rely on the SVD decomposition. Similarly, the alignment loss can very simply be adapted to the standard DFA (using $||W_i - B_i||^2$).
>
> ---
>
> While I like the idea of the paper, the obvious errors I have seen in the proof and in the first version make the rest of the paper very unreliable. The poor quality of some answers to my questions make me wonder whether the authors know what they are doing. The paper is not in a state where it should be accepted, so I am downgrading my score.
>
> That being said, the idea is indeed interesting and I encourage the authors to work on it and resubmit. I would advise dropping the proofs, which are not necessary to the rest of the paper, and focus more on the experimental part. I especially would like to see a fairer comparison with DFA, by either adding the alignment losses to DFA, or removing them from SSA. The current paper seems to try to do two things at once: i) making DFA more memory efficient using low-rank decomposition, and ii) improving DFA using new alignment losses. These are two very different contributions to the field, and should be made clearer -- it would also help selling the method better.

---

> > ### Author Response · Authors · 2024-12-03
> >
> > We sincerely thank the reviewer for their deep insights regarding our work. We have carefully considered their comments and will incorporate their suggestions into the next iteration of our paper. Below, we address each point in detail:
> >
> > 1. Theoretical Analysis and Independence of Local Losses:
> >    We have addressed the issue of local losses not being independent in Section A.2.3. Specifically, we state:
> >    "We assume that the linear separability condition holds for this convergence, which means that for early layers, the loss produced at each layer guides subsequent layers as well. However, from empirical results, we observe that for deeper networks (beyond ResNet-32), this assumption no longer holds."
> >    In future iterations, we will further clarify this statement and refine the associated theoretical analysis.
> >
> > 2. Extending Custom Loss to Standard DFA:
> >    We misunderstood the reviewer’s suggestion regarding extending custom loss components to standard DFA. We interpreted it as applying all components of the custom loss rather than individual components. We apologize for this miscommunication and will revisit this idea for more focused comparisons.
> >
> > 3. Orthogonal Loss Function and Smoothness:
> >    For the orthogonal loss, we emphasize that the SVD components of the weights need to be bounded to approach smoothness. This constraint ensures better theoretical grounding. Additionally, we clarify that while we formulated our proofs independently, we used large language models (LLMs) to verify mathematical formulas and improve the clarity of our proofs.
> >
> > 4. Convergence of DFA with Additional Losses:
> >    As there is no general proof of DFA’s convergence, it is challenging to derive theoretical insights into the convergence of the model with added losses, as stated by the reviewer. However, our approach focuses on making the composite loss function more convex, aiming to improve the convexity of the overall model function. We acknowledge that this assumption fails for deeper networks (beyond ResNet-32) and plan to address this limitation in future work.
> >
> > We greatly appreciate the reviewer’s feedback, which has helped us identify key areas for improvement and refinement.

---

### Official Review · Reviewer_SmsB · 2024-10-29

**Soundness:** 2
**Presentation:** 1
**Contribution:** 2
**Rating:** 3
**Confidence:** 5

**Summary:**

This paper proposes SVD-Space Alignment (SSA), a local layerwise training method combining Direct Feedback Alignment (DFA) and Singular Value Decomposition (SVD) to layer weights.
The authors propose to decompose the weights and feedback matrices using SVD and to use heavily regularized local losses to ensure alignment, sparsity and orthogonality.

Experiments on three networks of different depths are conducted, presenting competitive results with back-propagation on image classification tasks with a significant reduction of memory usage and computational cost during training.

**Strengths:**

1. Using low rank factorization together with DFA is an interesting and natural idea.

2. The presented results seem competitive with back-propagation (BP), while heavily reducing memory usage and computational cost during training.

3. An interesting ablation study over each component of the composite loss is provided.

**Weaknesses:**

1. A major issue of the paper is the lack of coherence of the writing and somewhat evident mistakes.
\
Equation 6 for example states that the cross entropy loss is: $L_{CE}=y_{predict}-y_{label}$, while this is instead supposed to be its derivative with respect to the predictions.
\
In standard DFA this derivative is denoted $e$ and is then projected onto every layer $l$ thanks to fixed feedback matrices $B_l$, with a weight update reading: $W_l^{t+1}=W_l^t - \eta B_l e \odot f'(a_l)h_{l-1}^T$, with $a_l$ and $h_l$ being respectively the pre-activation and activation of the layer.
The choice of the loss to optimize in DFA (and thus of its derivative $e$) is thus left to the user, which is clearly not the case in the paper.

2. The notations should be revised in order to be coherent.

3. The composite loss comprises some terms that are not well motivated nor explained. While the provided ablation study is interesting, it clearly lacks details as for example one could ask if the cosine similarity loss indeed improves the cosine similarity. The behavior of each individual objective with respect to the others could also be interesting to study.

4. Some sentences sound like claims and are not motivated: eg l.254: what are the "stable and smooth updates" ensured by DFA? Is there a study of the "stability of the training process" (l.262)?

5. The empirical results are somehow strange in many ways:
- one would expect a more extensive comparison with DFA to be conducted, as SSA is basically a low-rank approximation of DFA
- As DFA has been successfully extended to ResNet-56 by (Sanfiz and Akrout, 2021) with open-sourced code and even to Transformers by (Launay et al., 2020), excluding DFA evaluation from Table 2 where shallower networks (VGG-13 and ResNet-32) are tested claiming it is unable to scale to larger networks is false.
- The reported results with PEPITA on a 3 convolutional layers networks (Table 1) are very surprising as the original paper by (Dellaferrera and Kreiman, 2022) only reported results for one convolution and no other more recent paper has successfully trained a deeper convolutional network to the best of my knowledge. I would be very interested to know more about how those results were obtained as it has been observed by (Srinivasan et al., 2024) that PEPITA gets progressively worse results as the network grows deeper (for MLPs). I would expect that this observation would stay true for convolutional networks but the results seem to be the exact same as for the architecture used in the original paper by (Dellaferrera and Kreiman, 2022)
- I find the reported results on ImageNet very strange as the Top-5 Accuracy is lower than the Top-1 Accuracy (Table 2). Are the two column titles inverted?

6. The theoretical analysis provided in the appendix is unfortunately false as the composite loss function is not Lipschitz smooth as the cosine similarity component is not Lipschitz smooth. Linee 830-834 are false if $x$ is very small.

7. The paper would greatly benefit from a thorough proofread.

**Questions:**

8. What are your interpretation of the ranges of the different hyperparameters of the composite loss, given that some components are bounded and some are not?

9. How would different scheme of initializing the feedback matrices could impact the results you give as two components of the composite loss are dependent on those matrices?

10. How do you perform SVD on convolutional layers?

11. How do you update specific operations in ResNets such as downsampling convolutions, batch-normalization, etc?

---

> ### Author Response · Authors · 2024-11-23
> **Response to Reviewer pt 1**
>
> We thank the reviewer for taking the time to write a detailed feedback.
>
> We answer the questions here:
> i) Equation 6 issues. (Weaknesses pt 1 and 2):
>
> We appreciate the feedback. We indeed made some mistakes in the notations, and have corrected in the revised paper (Section 3.3). We also updated Fig 1 for better understanding of the notations and gradient input to the layers.
>
> The local loss is essentially a culmination of cross-entropy loss, alignment loss, cosine loss, singular vector orthogonality regularizer and hoyer regularizer. We input the derivative of this loss (gradient) to the layers, where the feedback error (DFA) is essentially the gradient of cross entropy loss.
>
> ii) The composite loss comprises some terms that are not well motivated nor explained. While the provided ablation study is interesting, it clearly lacks details as for example one could ask if the cosine similarity loss indeed improves the cosine similarity. The behavior of each individual objective with respect to the others could also be interesting to study.
>
> Authors: We have expanded on the role of individual loss components in Section 5.3 Ablation Study to provide greater clarity. The ablation results demonstrate that each loss term is critical for SSA's performance and efficiency. For example, Cross-Entropy Loss is essential for accuracy, while Cosine Similarity and Alignment Losses work together to preserve gradient direction in the absence of BP gradients. The Orthogonality Regularizer ensures meaningful angular transformations by maintaining the unitary properties of SVD components, and the Hoyer Regularizer sparsifies weights during rank reduction. While analyzing the interactions between loss components is beyond the current scope, we recognize its importance and have included it as future work.
>
> iii) Some sentences sound like claims and are not motivated: eg l.254: what are the "stable and smooth updates" ensured by DFA? Is there a study of the "stability of the training process" (l.262)?
>
> Authors: We have revised our paper accordingly to refine our motivations. We only show stability in our theoretical analysis and through convergence in our empirical results. Our loss components ensure that the gradient direction is maintained like BP (cosine-similarity), the unitary properties of SVD-components are maintained (orthogonality regularizer) so that any angular or linear transformation on those components will preserve the angles in the components. The alignment loss further tries to align the weight-space for better gradient direction updates. The feedback error essentially projects the global loss gradient into the layers to optimize global loss in a certain way. We account for the stability through the meaningful local loss optimization.
>
> iv) The empirical results are somehow strange in many ways (Weaknesses pt 5)
>
> Authors: We appreciate the reviewer’s observations and have addressed these points as follows:
>
> SSA vs. DFA Comparison:
> We have added a detailed discussion of SSA versus DFA and its variants in Section 6.1. While DFA can be extended to deeper networks such as ResNet-56, it suffers from significant accuracy loss. This is what we intended by stating that DFA is not easily extensible to larger networks. Our aim was to highlight SSA’s ability to mitigate such issues through low-rank approximations and structured local updates.
>
> PEPITA Results:
> The reported PEPITA results are sourced from the work by Apolinario et al. (2024): "LLS: Local Learning Rule for Deep Neural Networks Inspired by Neural Activity Synchronization" (arXiv:2405.15868). This study successfully trained PEPITA on networks deeper than those reported in the original paper by Dellaferrera and Kreiman (2022). We have cited this work in our revised manuscript for clarity.
>
> ImageNet Results:
> The reviewer is correct in noting the inconsistency between Top-1 and Top-5 accuracy in Table 2. This was a typographical error, and the column titles were inadvertently swapped. We have corrected this error in the revised version of the paper.
>
> v) The theoretical analysis provided in the appendix is unfortunately false as the composite loss function is not Lipschitz smooth as the cosine similarity component is not Lipschitz smooth. Linee 830-834 are false if x
>  is very small.
>
> Authors: We appreciate the reviewer’s observation regarding the theoretical analysis. While it is true that the cosine similarity component is not inherently Lipschitz smooth, we apply constraints to ensure bounded gradients. Specifically, we project the non-convex components, such as the cosine similarity loss, onto a convex sublevel set by normalizing the norms of
> x and y to 1. This ensures that the gradient differences are bounded, making the quasi-convex cosine similarity loss Lipschitz smooth under these constraints. We apply these constraints in our custom loss function as well.

---

> > ### Author Response · Authors · 2024-11-23
> > **Response to Reviewer pt 2**
> >
> > vi) What are your interpretation of the ranges of the different hyperparameters of the composite loss, given that some components are bounded and some are not?
> >
> > Authors: We have added our interpretation of the hyperparameter ranges in Section 4 Hyperparameter Selection.  Higher weights are assigned to convex components like feedback cross-entropy and alignment loss, and lower weights are kept for non-convex terms(quasi-convex terms) such as cosine similarity, orthogonality, and the Hoyer regularizer. To address non-convexity, we project cosine similarity and SVD component norms onto the unit sphere and smoothen the Hoyer regularizer (\ref{smooth_hoyer}), applying it only during rank reduction epochs.
> >
> > Theoretical analysis and Section 5.3 Ablation Study guide these choices, and
> > k=3-fold cross-validation on a small grid confirms their robustness with minimal overhead (5%-10% of training time). Once selected, the coefficients are fixed across experiments.
> >
> > vii) How would different scheme of initializing the feedback matrices could impact the results you give as two components of the composite loss are dependent on those matrices?
> >
> > Authors: We normally use Xavier initialization for our layers. To ensure that our matrices are unitary from the start, we first multiply the B_U,B_S,B_Vt for our feedback, then perform svd on the feedback matrix again, and reinitialize the svd decomposed feedback layers. So, different scheme of initializing feedback matrices doesn't impact the results as much.
> >
> > viii) How do you perform SVD on convolutional layers?
> >
> > Authors: A detailed explanation of SVD for convolutional layers is provided in Section 3.1 of the revised paper.
> >
> > Briefly, the convolutional kernel $K \in \mathbb{R}^{N \times C \times H \times W}$ is reshaped into a 2D matrix $K' \in \mathbb{R}^{NW \times CH}$ and decomposed using SVD:
> > $
> > K' = U \Sigma V^T,
> > $
> > where $U$, $\Sigma$, and $V^T$ represent the singular vectors and singular values, with $r = \min(NW, CH)$. The components are then reshaped into two convolutional kernels:
> >
> > $U \sqrt{\Sigma}$ becomes $K_1 \in \mathbb{R}^{r \times C \times H \times 1}$,
> >
> > $\sqrt{\Sigma} V^T$ becomes $K_2 \in \mathbb{R}^{N \times r \times 1 \times W}$.
> >
> > This decomposition replaces the original convolution with two consecutive operations, preserving spatial and hierarchical features while reducing computational complexity. During backpropagation, gradients are directly computed for $K_1$ and $K_2$, and the decomposed kernels are retained for inference.
> >
> > ix) How do you update specific operations in ResNets such as downsampling convolutions, batch-normalization, etc?
> >
> > Authors: We have provided the details for it in the updated Appendix A.3.
> >
> > We hope that these revisions and clarifications adequately address the reviewer’s comments. We thank the reviewer for their insightful comments.

---

> > > ### Comment · Reviewer_SmsB · 2024-12-02
> > >
> > > I thank the authors for taking into account my remarks. I thoroughly re-read the improved paper.
> > > Overall the paper is much improved, unfortunately, it still requires a lot of improvement to be accepted, especially in the experimental part.
> > >
> > > Let me point out some incoherences:
> > > - Table 1 seems very weird:
> > > >DFA was not originated by (Akrout et al., 2019). Furthermore this specific paper focused on learning the feedback connections in FA setting and not DFA. Lastly the results they got was on par with BP for ResNet 18 and ResNet 50 when trained on ImageNet.
> > >
> > > > The PEPITA results you mention come from a recent preprint paper. This preprint reports false results on PEPITA, claiming training on a SmallConv network but reporting **the exact same results** as the original PEPITA paper on a single convolutional network. The code attached in this preprint futher does not include the PEPITA baseline.
> > >
> > > I would like to outline that the authors should not base themselves on preprints to copy/paste some results and propagate false information. The baselines should be sourced from published papers or (in the ideal case) re-runned to verify coherence.
> > > The given results should also be attributed to the correct paper as (Akrout et al.) did not report results for DFA.
> > > These errors make the whole experimental section unreliable.
> > >
> > > - There is no hyper-parameter search for the composite loss reported. A 5 (!) component loss is presented with fixed hyper-parameters set across the experiments. Though a discussion is now in appendix, the choice of these hyper-parameters still seems coming out of pure luck.

---

> ### Author Response · Authors · 2024-12-03
>
> We thank the reviewer for highlighting the incoherences in our work.
>
> i) Regarding Table 1: The DFA results were mistakenly attributed to Akrout et al., 2019. This was a typographical error, and we sincerely apologize for the oversight.
>
> ii) Regarding the Preprint: Upon contacting the authors of the preprint, they confirmed that PEPITA was not run on SmallConv, as reported. Instead, the results are based on the same architecture used in the original PEPITA paper.
>
> iii) We have addressed our limitation of the hyperparameter search for the loss coefficients in the paper. We will further deep dive in the limitation in our future work.
>
> We acknowledge these errors and will ensure they are corrected in the next iteration of our paper. Thank you for bringing these issues to our attention.

---

### Official Review · Reviewer_8tHJ · 2024-11-03

**Soundness:** 3
**Presentation:** 3
**Contribution:** 3
**Rating:** 8
**Confidence:** 3

**Summary:**

This paper proposes a backpropagation-free training framework called SVD-Space Alignment (SSA). SSA decomposes weight matrices by Singular Value Decomposition (SVD) and then updates the SVD components by direct feedback alignment (DFA) under the guidance of customized layerwise loss fuctions. The experimental results demonstrate that SSA achieves classification accuracy close to that of backpropagation however with significantly reduced memory usage, computational cost, and energy consumption. A theoretical proof for SSA convergence guarantees is also presented. This novel local training framework provides a promising energy and computation-efficient solution for deep learning in resource-constrained situations.

**Strengths:**

The paper creatively combines SVD and DFA for neural network training. The authors introduce a tailored layerwise loss function that incorporates several different constraints so that the local layerwise updates during training are both convergent and efficient.

As network sizes grow, training becomes increasingly unaffordable for organizations and individuals with limited resources. This research presents a compelling alternative to conventional backpropagation for training deep neural network models.

This paper is well organized and clearly written.

**Weaknesses:**

The effectiveness of SSA for training deep convolutional neural networks has been validated in this work. Given that transformers currently dominate in domains such as Natural Language Processing (NLP), it would be beneficial for future research to explore the extension of SSA to other architectures.

**Questions:**

1. Beyond theoretical analysis, it would be beneficial to present empirical results that demonstrate the convergence rate and training stability of SSA in comparison to BP.

2. A dynamic rank reduction strategy is used to reduce the rank of weight matrix progressively. How is it implemented and guranteed, especially considering that updates to SVD components are not inherently directed towards reducing rank.

3. The decomposition of a convolutional layer while retaining its hierarchical information is described in Appendix A.3, however it may be challenging for readers to grasp. Including a schematic diagram would enhance understanding by providing a visual aid.

---

> ### Author Response · Authors · 2024-11-23
> **Response to Reviewer**
>
> We thank the reviewer for highlighting the strengths of our paper. Below, we provide detailed answers to the reviewer's concerns.
>
> i) A dynamic rank reduction strategy is used to reduce the rank of weight matrix progressively. How is it implemented and guranteed, especially considering that updates to SVD components are not inherently directed towards reducing rank.
>
> Authors: We have updated the section 3.4 to include detailed explanation of how we reduce the rank of weight matrix progressively. Every ten epochs, we apply Hoyer regularizer as a part of our custom loss objective to increase the sparsity (ratio of L1 norm of singular values to L2 norm of singular values). This reduces rank. We also make threshold based checks (check if the rank of the weight matrix is below 95\% of the matrix's energy) in the later part of the epochs, to stop aggressive reduction in rank, losing a part of the representational capacity. If we see that the rank reduction is not effective based on the threshold based check, we halt the epoch-based scheduling. In the initial epochs, we can reduce rank without worrying for reduction in representational capacity, as then the network can update itself for the change in the ranks in the next epochs.
>
> ii) Beyond theoretical analysis, it would be beneficial to present empirical results that demonstrate the convergence rate and training stability of SSA in comparison to BP.
>
> Authors: We provide error convergence graphs for a small DNN in Section 6.1 in the updated paper.
>
> iii) The decomposition of a convolutional layer while retaining its hierarchical information is described in Appendix A.3, however it may be challenging for readers to grasp. Including a schematic diagram would enhance understanding by providing a visual aid.
>
> Authors: We have put a detailed explanation of the convolutional layer decomposition in SVD-space in Section 3.1, as per the suggestions.
>
> We hope these updates address the reviewer’s concerns and welcome any additional suggestions or clarifications.

---

### Official Review · Reviewer_Rq2e · 2024-11-03

**Soundness:** 2
**Presentation:** 2
**Contribution:** 2
**Rating:** 3
**Confidence:** 3

**Summary:**

The paper proposes a new learning framework dubbed SSA that combines DFA with SVD. A local loss function with five components and a dynamic rank reduction strategy are adopted to train accurate and efficient DNNs. Comprehensive experimental results demonstrate that SSA can reach comparable accuracy performance as BP while reducing memory and computational cost by a large margin.

**Strengths:**

1. The idea of combining DFA with SVD is novel.
2. The designed loss function takes care of many different aspects of training.

**Weaknesses:**

1. The paper focuses on reducing memory and computational cost during training. However, to deploy models in resource-constrained environments, efficiency during post-training inferences is more important.
2. It's better to provide comparison between the proposed method and model compression techiques like quantization-aware training (QAT).
3. The proposed method has many limitations such as hyperparameter sensitivity and inability to scale to larger models. Also, no evidence is provided to show the effectiveness of SSA on transformer-based models.

**Questions:**

Please see the weakness part.

---

> ### Author Response · Authors · 2024-11-23
> **Response to Reviewer**
>
> We thank the reviewer for their time and consideration for reading our paper.
>
> We answer your queries here:
>
> i) The paper focuses on reducing memory and computational cost during training. However, to deploy models in resource-constrained environments, efficiency during post-training inferences is more important.
>
> Authors: We have revised Section 3.5 to demonstrate both training and inference complexity. The computational complexity of inference is reduced to:
> $O(r \times n) + O(r \times r) + O(m \times r)$
> where $r \ll \min(m, n)$. This decomposition not only reduces compute cost but also minimizes memory requirements, as only the SVD components ($U$, $S$, $V^T$) need to be stored:
> $
> O(m \times r) + O(r \times r) + O(r \times n).
> $
> By dynamically reducing the rank $r$ during training, SSA further optimizes both compute and memory usage in the inference phase, making it highly suitable for deployment in resource-constrained environments such as edge devices.
>
> ii) It's better to provide comparison between the proposed method and model compression techniques like quantization-aware training (QAT).
>
> Authors: We provide the following tables highlighting the differences between SSA and QAT, and then compression percentages for QAT and SSA methods.
>
> | **Aspect**               | **SSA (Ours)**                 | **QAT**                        |
> |---------------------------|---------------------------------|---------------------------------|
> | **Objective**            | Low-rank training optimization | Precision-aware training       |
> | **Application Stage**    | Full training and inference    | Full training and inference    |
> | **Efficiency Gains**     | Memory and compute savings     | Reduced bit-width (e.g., 8-bit)|
> | **Training Scope**       | Low-rank updates to weights    | Quantized forward and backward passes |
> | **Accuracy Impact**      | Comparable to BP               | Slight drop at low precision (e.g., 4-bit) |
> | **Flexibility**          | Adaptive to model size and rank| Fixed precision during training|
> | **Biological Plausibility** | Yes                         | No                             |
>
> We provide the comparison of comparison methods obn VGG-11 reflecting inference-time memory savings.
>
> | **Method**          | **Bit-Width (W/A) or Rank (r)** | **Compression (%)** | **Accuracy (%)**      |
> |----------------------|---------------------------------|----------------------|------------------------|
> | **Full-Precision**  | 32/32                          | 0%                  | 91.7 to 93.8          |
> | **BinaryConnect**   | 1/32                           | 48.44%              | 91.73                 |
> | **BNN**             | 1/1                            | 98.44%              | 89.85                 |
> | **HWGQ**            | 1/2                            | 97.66%              | 92.51                 |
> | **LQ-Nets (3/2)**   | 3/2                            | 95.31%              | 93.8                  |
> | **DMBQ**            | 0.7/32                         | 48.91%              | 93.7                  |
> | **SSA (Ours)**      | Rank-reduced              | 75%                 | Comparable to BP      |
>
> Ref: Babak Rokh, Ali Azarpeyvand, and Alireza Khanteymoori. 2023. A Comprehensive Survey on Model Quantization for Deep Neural Networks in Image Classification. ACM Trans. Intell. Syst. Technol. 14, 6, Article 97 (December 2023), 50 pages. https://doi.org/10.1145/3623402
>
> iii) The proposed method has many limitations such as hyperparameter sensitivity and inability to scale to larger models. Also, no evidence is provided to show the effectiveness of SSA on transformer-based models.
>
> We have addressed hyperparameter settings in Section 4, Hyperparameter Selection. We state the hyperparameters used in that section based on the constraints of our theroretical analysis and verifying them in k=3-fold cross-validation. While our paper does not focus on exhaustive hyperparameter tuning, we acknowledge this as a potential limitation and have discussed it in the Limitations section. Future work may explore more sophisticated methods for tuning these parameters.
>
> We can scale to ResNet-32 and convolutional layers, which is not the case for most feedback alignment-based local learning methods. Regarding transformer-based models, the scope of this work was primarily to validate a novel local learning rule in the SVD-space of DNN layers. While our experiments focus on CNNs, the design of SSA, particularly its operation in the SVD-space and optimization of local loss objectives, is architecture-agnostic. Future research will extend SSA to deeper networks and to transformer-based models, leveraging their modular self-attention mechanisms and natural compatibility with our rank-reduction strategy.
>
> We thank the reviewer for prompting us to consider the efficiency of post-training inferences more deeply and to further explore the compression aspect of our lightweight model.

---

### Official Review · Reviewer_HRY5 · 2024-11-03

**Soundness:** 2
**Presentation:** 2
**Contribution:** 2
**Rating:** 5
**Confidence:** 3

**Summary:**

Motivated by reducing the memory and computational cost of training deep neural networks, the authors propose a local training methodology based on Direct Feedback Alignment (DFA). DFA is limited in its application due to poor scaling to deep models and complex tasks. The authors propose to improve upon DFA by decomposing the model's weight matrices into orthogonal components using SVD, perform local updates on the orthogonal components, and to decrease the rank of this representation over the course of training. The authors provide convergence proofs in the appendix, and evaluate their method in training VGG-13, ResNet-32, and a custom small 3-layer CNN on CIFAR-10, CIFAR-100 and ImageNet.

**Strengths:**

* The methodology of the proposed extension of DFA proposed is intuitive, and interesting, specifically having a decreasing rank schedule over training and decomposition of the weight matrices.
* The method is well-motivated with compute and memory complexity being a challenging in contemporary applications and research with deep neural networks.
* The related-work does mention many other local training methods, although it might be a bit too dismissive of many as baselines.
* The authors provide a small ablation over the five different loss components, although this could have been more detailed.
* Please note I did not evaluate the convergence proofs in the appendix, as they were not in the main paper. Given the 5 different loss components in the methodology proposed however, I do believe they are necessary.
* Both top-1 and top-5 are evaluated for Imagenet (unlike so much contemporary work)

**Weaknesses:**

* Despite the main motivation of methodology/paper, only real-world compute/memory complexity results in paper (including appendix) are VGG-13/Figure 2. This is very far from sufficient, and VGG itself is not an architecture that should be used when claiming efficiency results to begin with. The authors have accuracy results from more reasonable architectures (e.g. ResNet-32) with their method, so there is no reason not to include the compute/memory savings also. Notably the authors are themselves aware that lightweight architectures (e.g. MobileNet in 6.1) would be much better motivated, so I'm not seeing good reasons for the focus on only VGG.
* No variance/stddev in results. The results are lacking any mention of having been evaluated over multiple random inits, etc, and no variance or other measure of significance of the results is available. For small datasets/models, and with the claims behind the method being computationally efficient, there is no reason not to evaluate over multiple training runs and provide mean/variance over these runs to better evaluate the generalization results.
* Although hard to evaluate due to the above, with the results as presented, there is not clear evidence that the results are significant. In many of the results AugLocal or SVD-BP appear to be very close or better than the results of SSA.
* The work is not repeatable as presented, with a lack of experimental details, although there are some in appendix:
    * Experimental details are pronounced generally, e.g. "learning rates ranging from 1e-4 to 5e-4", rather than listed for specific experiments.
    * The five loss coefficient hyper-parameters ($\alpha, \beta, \gamma,\delta,\epsilon$), are very important. In the appendix the values for these hyper-parameters are listed to be used "ideally", and I'm not sure what this means, does it means they vary or not? Unfortunately there is both very little explanation as to how those are found (except "cross-validation"), the exact values of these hyper parameters used for each of the experimental results themselves (unless they are all the same).
     * Another example of the methodology I would consider a hyper parameter, the choice of which is poorly explained, is the rank reduction schedule.
     * Missing experimental details for CIFAR-100.
     * Missing experimental details for ImageNet.
* Tables are really all over the place, often with a single odd line of text randomly interspersed between them, making it hard to read both the paper and the tables - must have been forced with [H]. Recommend authors use [tbp] float placement for all their tables to fix this.
* In 6.2 (limitations) to their credit, the authors explain that the method is very sensitive in particular to the five loss coefficient hyper-parameters ($\alpha, \beta, \gamma,\delta,\epsilon$), as might be expected. Anyone who has attempted to work with a loss with even a few coefficients can recognize how unstable/hard such a methodology might be in practice. It's hard from the paper (as-is) to judge if using SSA requires these needed to be changed for each experiment or were kept constant for different models/datasets. If it's the former, it would be hard to justify the method as reducing training times in practice.

**Questions:**

* What are the real-world compute/memory results for the other models in your work?
* How exactly did you find your hyper parameters, and what are the hyper-parameters (and all other experimental details required to repeat the experiments)?
* How much do the five loss coefficients themselves have to be tuned to get good performance, i.e. how different are they for your different models/results? This is important as if they have to be found for each different experimental setup by sweeping training, it's hard to justify the method as speeding up training.
* Why did you choose to demonstrate improvements in training compute/memory usage not on a lightweight architecture, but instead VGG which is not used anymore in research/applications and widely recognized to be highly inefficient compared to e.g. ResNets, MobileNet, EfficientNet, etc.

---

> ### Author Response · Authors · 2024-11-23
> **Response to Reviewer**
>
> We thank the reviewer for their detailed and valuable feedback, which led to huge improvements in the revised paper.
>
> We address each of your comments below:
>
> From Weaknesses and Questions:
>
> i) Despite the main motivation of methodology/paper, only real-world compute/memory complexity results in paper (including appendix) are VGG-13/Figure 2.
> and
> What are the real-world compute/memory results for the other models in your work?
>
> Authors: We have incorporated the compute and memory savings graphs for MobileNetV1 and ResNet-32, alongside VGG-13, as part of Figure 2 in Section 5.2, in our revised paper. We thank the reviewer for this insight.
>
> ii) No variance/stddev in results.
>
> Authors: We have updated Table 1 in Section 5.2 to include variance and standard deviation in the results in our revised paper.
>
> iii) Although hard to evaluate due to the above, with the results as presented, there is not clear evidence that the results are significant. In many of the results AugLocal or SVD-BP appear to be very close or better than the results of SSA.
>
> Authors: We appreciate the reviewer’s comment. BP and SVD-BP rely on finely tuned gradient-based optimization and a global loss signal, enabling strong layer-to-layer coordination. In contrast, local learning methods, driven by localized objectives, inherently lack such coordination. Achieving competitive results with BP using a local learning approach, as SSA does, is a significant milestone and remains an active area of research.  Also, SVD-BP has to store gradients as BP and requires more memory and compute. AugLocal (layer-wise training) introduces additional overhead through auxiliary networks and classifiers (faring a bit better on deeper networks), SSA achieves competitive accuracy with significantly lower memory and compute requirements, making it a more efficient alternative.
>
> iv) The work is not repeatable as presented, with a lack of experimental details, although there are some in appendix.
> and
> How exactly did you find your hyper parameters, and what are the hyper-parameters (and all other experimental details required to repeat the experiments?
>
> Authors: We have made the following updates to address your concerns:
>
> Hyperparameter Selection:
> We have added a detailed paragraph in Section 4: Experimental Setup to explain our hyperparameter choices. Specifically, we used a learning rate of $1 \times 10^{-4}$ for smaller datasets (e.g., CIFAR-10) and $5 \times 10^{-4}$ for larger datasets (e.g., ImageNet). The loss based hyperparameter values were kept consistent across experiments, with adjustments made only to the overall learning rate for layerwise updates. As layer objectives are largely independent, tuning coefficients involves optimizing simpler, localized objectives, allowing faster cross-validation with a narrower range of candidate values.
>
> Rank Reduction Schedule:
> We expanded Section 3.4 to provide a clearer explanation of our progressive rank reduction strategy, highlighting how it balances memory and computational efficiency while maintaining accuracy.
>
> Experimental Details for CIFAR-100 and ImageNet:
> Detailed training setups for CIFAR-100 and ImageNet, including preprocessing, hyperparameter settings, and hardware specifications, have been added to Appendix A.3 to ensure transparency and reproducibility.
>
> v) How much do the five loss coefficients themselves have to be tuned to get good performance, i.e. how different are they for your different models/results? This is important as if they have to be found for each different experimental setup by sweeping training, it's hard to justify the method as speeding up training.
>
> Authors: In Section 4 of our revised paper, we describe our hyperparameter selection process. Coefficients were kept consistent across all experiments, with only the overall learning rate adjusted for layerwise updates. Robustness of the coefficients was verified through a small
> k=3-fold cross-validation, taking 5–10% of training time. While layerwise tuning is simpler and faster than global loss tuning (due to simpler layerwise loss objectives), we acknowledge that this paper does not extensively explore optimal hyperparameter tuning, as we have mentioned in the paper in Section 6.2 limitations.
>
> vi) Why did you choose to demonstrate improvements in training compute/memory usage not on a lightweight architecture, but instead VGG which is not used anymore in research/applications and widely recognized to be highly inefficient compared to e.g. ResNets, MobileNet, EfficientNet, etc.
>
> Authors: There was no particular reason to choose VGG, we chose it due to the ease of visibility of the results in the layers.
>
> Once again, thank you for your time and consideration, and for giving us such a detailed review with the opportunity to improve our paper.

---

> > ### Comment · Reviewer_HRY5 · 2024-11-30
> >
> > First I want to thank the authors for their rebuttal and work in updating the paper, and addressing many weaknesses I identified. I also want to apologize for my late reply --- I did write this comment earlier, but unfortunately it was lost by open review before I could post it, and I didn't have enough time to rewrite it until now:
> >
> > Quickly going through the revised paper it's obvious that:
> > - Figures are much improved, are now readable, along with many (although not all) figure captions also being improved.
> > - Computational complexity and memory usage for a range of much more reasonable models are presented, notably in Figure 2.
> > - Many more experimental details now included, making the work more repeatable and understandable.
> > - Table 1 now has variance/mean.
> >
> > Remaining issues:
> > - The computational complexity and memory usage appear to be theoretical only, which is problematic in evaluating the method's real-world impact. Theoretical FLOPS (if that's what this is, as it's not clear) do not reflect the real-world performance of algorithms, especially when on GPU.
> > - All results except for perhaps very computationally expensive experiments should have mean/variance or some idea of significance. ImageNet/ResNet 50 even is not that expensive to run with 5 different seeds on today's GPU hardware.
> > - ImageNet/ResNet32 - ResNet 32 is an architecture built for CIFAR-10, not ImageNet. Not clear to me at all how the architecture is being changed to run with ImageNet/how this is a good model to be used with ImageNet.
> > - Having read the other reviewer's concerns on the comparison with DFA in particular, I'm a bit concerned with this myself now, and am hoping to see their comments on the rebuttal.
> > - I'm still worried about the hyperparameter selection for the loss coefficients, and the author's rebuttal didn't really address my concerns on how sensitive these are, or when they need to be changed in practice.
> >
> > Further more minor notes for improvement:
> > - Figure axes need to be labelled with units where appropriate (e.g. in Fig 2, not clear what they are)
> > - Table captions should be on top of the table (unlike figure)
> > - It's hard for me to tell exactly what changed in the paper, you might want to highlight differences in the text with a different colour in future.
> >
> > Summary:
> > Overall the paper is much improved, and I will revise my rating considering those changes. Unfortunately, I don't think the paper is at a point where it should be accepted yet, I believe there is a lot more to do still. I am hopeful that the authors are able to present more convincing real-world benchmarks in future iterations of the paper, along with significant revisions in the writing and presentations of figures/tables, and would encourage them to do so.

---

> ### Author Response · Authors · 2024-12-01
>
> I thank the reviewer for their concerns. We would address the concerns in our future work or the next iterations.
>
> i) ImageNet/ResNet32 - ResNet 32 is an architecture built for CIFAR-10, not ImageNet. Not clear to me at all how the architecture is being changed to run with ImageNet/how this is a good model to be used with ImageNet.
>
> Authors: We appreciate the reviewer’s observation regarding the use of ResNet-32 for ImageNet. We acknowledge that ResNet-32 is traditionally designed for CIFAR-10 and not directly optimized for ImageNet. In our experiments, we adapted ResNet-32 for ImageNet by increasing the input resolution, modifying the stride and pooling layers, and adjusting the number of filters to handle the larger image sizes and dataset complexity. As our model can't yet extend to ResNet-50 (due to linear separability as we had mentioned in limitations) , we had adapted ResNet-32 for ImageNet. Local training methods usually suffer for deeper network layers (beyond 10 layers). We are keeping the extension of our model to deeper networks for our future work.
>
> For the other improvements suggested, we will include those improvements in our paper in the next iteration.
>
> I thank the reviewer again for contributing to the improvisation of our paper.

---

### Author Response · Authors · 2024-11-28
**Thank you to all the reviewers; Summary of major changes**

To all the reviewers,

Thank you so much for your valuable comments. We got some great questions that helped us improve our manuscript, both in experimentation and presentation.

Here are the major changes made to the revised manuscript:

i) Figure 1 is updated with the updated notations which leads to clearer and better understanding. Also, we have incorporated clearer and corrected understanding of the custom loss function in Section 3.3.

ii) We added the explanation for SVD-decomposition for convolutional layers in Section 3.1.

iii) We have added further details on dynamic rank reduction strategy in Section 3.4, and introduced a paragraph 'Hyperparameter Selection' in Section 4 Experimental Setup.

iv) We have added section 6.1 'Comparison with DFA' to our paper.

v) Lastly, we have revised our comments on Global Convergence of Model Loss in Section A.2.3.

Please let us know your comments on our response and revised manuscript. We are open to further discussion if needed.

---

### Meta-Review · Area_Chair_bCH2 · 2024-12-20

**Metareview:**

This paper presents a local training framework that replaces backpropagation in Deep Neural Networks (DNNs) to reduce memory and computational demands. By leveraging Singular Value Decomposition (SVD) and Direct Feedback Alignment (DFA), the method updates SVD components locally, integrating custom loss functions, regularization, and sparsity. It achieves comparable accuracy to backpropagation while reducing memory and computational costs by 50–75%. The authors claim that their framework offers an efficient, scalable solution for deep learning in resource-constrained settings, with theoretical convergence guarantees and publicly available code.

Based on the reviews the strengths and weaknesses of the paper are as follows:

Pros:

+ The combination of Singular Value Decomposition (SVD) with Direct Feedback Alignment (DFA) seems to be novel idea for local training frameworks.

+ The method significantly reduces memory and computational costs during training, addressing critical resource constraints.

+ Achieves classification accuracy competitive with backpropagation (BP) while being more efficient.

+ Theoretical Contribution: Includes convergence guarantees and a dynamic rank reduction strategy to optimize model complexity.

Cons:

- Experimental Gaps: Limited evaluation on architectures beyond CNNs; lacks exploration of transformers or broader tasks.

- Hyperparameter Sensitivity: The composite loss introduces five hyperparameters, with insufficient details on tuning or generalizability.

- Lack of Repeatability: Insufficient implementation details and reliance on theoretical complexity without empirical validation of real-world performance.

- Comparison Issues: Limited and potentially biased comparisons to DFA and other methods like QAT; reliance on preprint results questioned.

- Theoretical Concerns: The claim of Lipschitz smoothness for the composite loss function is incorrect, undermining some theoretical proofs.

- Presentation Weaknesses: Coherence in writing, notations, and experimental reporting remain problematic despite improvements.

Following feedback, the authors improved the clarity of notations, experimental details, and added ablations demonstrating the impact of loss components. This helped resolve some of the concerns raised above. All reviewers agreed that some of the concerns were not addressed (including the first 4 mentioned above based on my understanding) and in fact the most positive reviewer said they would decrease their score (it doesn't seem like they have but clearly they indicated that they will reduce their score to 6). I agree and can not recommend acceptance at this time.

**Additional Comments On Reviewer Discussion:**

Following feedback, the authors improved the clarity of notations, experimental details, and added ablations demonstrating the impact of loss components. This helped resolve some of the concerns raised above. All reviewers agreed that some of the concerns were not addressed (including the first 4 mentioned above based on my understanding) and in fact the most positive reviewer said they would decrease their score (it doesn't seem like they have but clearly they indicated that they will reduce their score to 6). I agree and can not recommend acceptance at this time.

---

### Decision · Program_Chairs · 2025-01-22

Reject